# 3D Gaussian Splatting based Scene-independent Camera Relocalization with Unidirectional and Bidirectional Feature Fusion

**Junyi Wang**[1,2]
junyiwang@sdu.edu.cn

**Yuze Wang**[2]
wangyuze19980709@163.com

**Wantong Duan**[2]
wantongd@buaa.edu.cn

**Meng Wang**[2]
wangm05@buaa.edu.cn

**Yue Qi**[2,3*]
qy@buaa.edu.cn

## Abstract

Visual localization is a critical component across various domains. The recent emergence of novel scene representations, such as 3D Gaussian Splatting (3D GS), introduces new opportunities for advancing localization pipelines. In this paper, we propose a novel 3D GS-based framework for RGB based, scene-independent camera relocalization, with three main contributions. First, we design a two-stage pipeline with fully exploiting 3D GS. The pipeline consists of an initial stage, which utilizes 2D-3D correspondences between image pixels and 3D Gaussians, followed by pose refinement using the rendered image by 3D GS. Second, we introduce a 3D GS based Relocalization Network, termed GS-RelocNet, to establish correspondences for initial camera pose estimation. Additionally, we present a refinement network that further optimizes the camera pose. Third, we propose a unidirectional 2D-3D feature fusion module and a bidirectional image feature fusion module, integrated into GS-RelocNet and the refinement network, respectively, to enhance feature sharing across the two stages. Experimental results on public 7 Scenes, Cambridge Landmarks, TUM RGB-D and Bonn demonstrate state-of-the-art performance. Furthermore, the beneficial effects of the two feature fusion modules and pose refinement are also highlighted. In summary, we believe that the proposed framework can be a novel universal localization pipeline for further research.

## 1 Introduction

Visual localization is considered a fundamental research problem in computer vision and is applied across a variety of scenarios, including Augmented Reality (AR), Mixed Reality (MR), robotics, and autonomous driving Wang et al. (2024c); Jia et al. (2024); Zhu et al. (2024). The primary function of a localization algorithm is to estimate the 6-DoF (Degrees of Freedom) camera pose within a target environment.

As current works cover, two main types of methods are investigated to achieve robust localization performance, including feature matching and geometry regression. Specially, feature matching methods employ either hand-crafted Liu et al. (2017) or learned features Sun et al. (2021) to establish pixel correspondences for localization, while geometry regression approaches train the deep network

---

[*]1. School of Computer Science and Technology, Shandong University, Qingdao, Shandong, China.
2. State Key Laboratory of Virtual Reality Technology and Systems, Beihang University, Beijing, China.
3. Qingdao Research Institute of Beihang University, Qingdao, Shandong, China.
Corresponding Author: Yue Qi.

39th Conference on Neural Information Processing Systems (NeurIPS 2025).

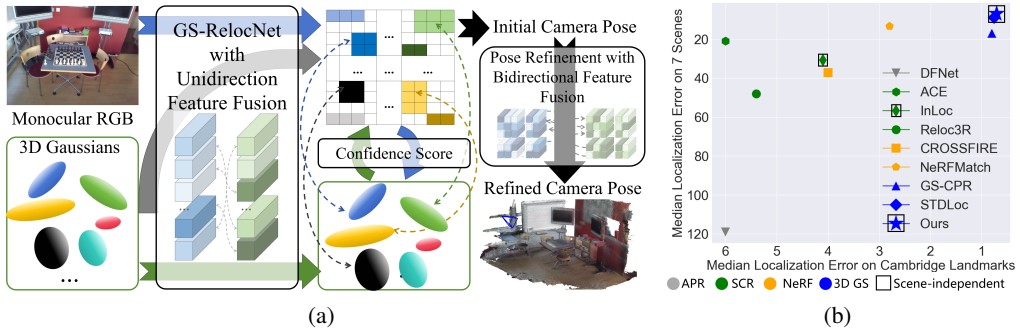

Figure 1: (a) Localization pipeline. The framework predicts the initial pose by establishing 2D-3D correspondences between pixels and 3D Gaussians, followed by a pose refinement process to optimize pose by rendered views using 3D GS. (b) Localization performance comparison. Our method obtains the least localization error on the two datasets, while supporting scene-independent localization.

to solve the camera pose, by Absolute Pose Regression (APR) Chen et al. (2022) or Scene Coordinate Regression (SCR) Wang & Qi (2023b). However, due to the representation, these methods always ignore texture and illumination information, which limits their capacity to fully represent the scene.

Recent neural and geometric 3D structures for Novel View Synthesis (NVS) have gained significant popularity for scene representatin Mildenhall et al. (2021); Kerbl et al. (2023); Hu et al. (2023). Specially, 3D Gaussian Splatting (3D GS) offers a well-balanced approach to training and rendering performance for NVS, presenting new opportunities for the localization pipeline. **However, how to leverage 3D GS for robust and accurate localization remains a significant challenge**.

Current methods predominantly utilize 3D GS for pose refinement Keetha et al. (2024); Yan et al. (2024); Liu et al. (2025), which heavily relies on the accuracy of initial camera pose estimation. When the initial pose estimation is inaccurate or fails, the refinement process becomes ineffective. To address this, **our motivation lies in the employment of 3D GS for both pose initialization and refinement, while achieving scene-independent relocalization to enhance robustness**.

As illustrated in Fig. 1(a), we introduce a novel relocalization framework that first establishes 2D-3D correspondences between image pixels and 3D Gaussians, followed by a refinement stage that predicts the relative pose between real and rendered views using 3D GS. A feature fusion module is incorporated in both stages to enhance correspondence regression. To the best of our knowledge, **this is the first 3D GS based, scene-independent relocalization framework**, offering a robust solution for challenging localization tasks. Specially, the term "scene-independent" indicates that our framework can achieve robust relocalization in a target scene without requiring scene-specific pre-training, in contrast to most "scene-dependent" methods that necessitate prior training on the target scene. Our contributions are summarized as follows.

1. We propose a innovative framework for scene-independent camera relocalization. The framework comprises initial pose estimation by establishing correspondences between pixels of the input image and the scene expressed by the 3D Gaussians, and pose refinement by predicting the relative pose between the input image and rendered view using 3D GS.

2. We design GS-RelocNet for predicting the correspondences to obtain the initial camera pose. Within GS-RelocNet, we introduce a unidirectional feature fusion module to merge geometry and texture features for learning confidence scores between each pixel and 3D Gaussian.

3. We propose a pose refinement network based on 3D GS. In the refinement network, we present a bidirectional feature fusion module to combine features from rendered and real images.

To validate the performance of the framework, we conduct experiments on 7 Scenes and Cambridge Landmarks. As illustrated in Fig. 1(b), the results demonstrate the state-of-the-art localization performance on the two datasets. Meanwhile, our framework can support scene-independent relocalization, denoting that it can perform relocalization in unseen scenes.

## 2 Related Works

### 2.1 Localization with Feature Matching

Traditional hand-crafted feature matching based methods typically follow a pipeline consisting of feature extraction, matching, global map construction, and optimization Liu et al. (2017); Sattler et al. (2017). Moreover, semantic SLAM systems Yang & Scherer (2019); You et al. (2023); Lin et al. (2024b); Xi et al. (2025); Zhang et al. (2025) incorporate semantic information derived from learned features to enhance the robustness and accuracy of traditional hand-crafted feature based processes. Alternatively, learned feature matching methods aim to estimation pixel correspondences for pose estimation Wang et al. (2022, 2024d). Based on LoFTR Sun et al. (2021), Efficient LoFTR Wang et al. (2024d) performed the transformer with an aggregated attention mechanism using adaptive token selection for efficiency.

### 2.2 Localization with Geometry Regression

The geometry regression methods can be broadly categorized into two approaches, including APR and SCR methods. APR methods train the deep network to learn the relationship between 2D images and 6-DoF camera poses Chen et al. (2022, 2024b). While APR methods offer high computational efficiency, they are often limited by accuracy and generalization issues Liu et al. (2024b). Alternatively, SCR techniques calibrate the pose by using the Kabsch or Perspective-n-Point (PnP) algorithm to the known source and evaluated target coordinate, which achieve considerable localization performance Wang & Qi (2021, 2023b). Recent hot SCR researches are DUSt3R Wang et al. (2024b) and its subsequent extension works Leroy et al. (2024), By using large training samples, the methods achieve outstanding localization accuracy and generalization ability. These methods focus on geometry regression, bug can not fully exploit texture features.

### 2.3 Localization with Neural Radiance Field (NeRF) and 3D GS

Due to the outstanding NVS performance, NeRF is applied to the localization pipeline with iterative rendering and pose updates Germain et al. (2022); Moreau et al. (2023); Chen et al. (2024a); Wang et al. (2023); Xu et al. (2024). NeRFect Match Zhou et al. (2024b) explored the potential of NeRF 's internal features in establishing precise 2D-3D matches for localization. With the shift in the NVS field from NeRF to 3D GS, STDLoc Huang et al. (2025) introduced a matching-oriented Gaussian sampling strategy and a scene-specific detector to achieve efficient and robust pose estimation.

## 3 Method

### 3.1 Overview

Given a target image and a 3D scene model expressed through 3D GS, our method predicts the 6-DoF camera pose of the target image within the scene. The overall localization process is composed of two stages, both utilizing 3D GS in distinct ways. In the initial pose estimation stage, we establish the correspondences between each image pixel and 3D Gaussian, followed by a PnP algorithm with RANSAC to solve the initial camera pose. Based on the predicted pose, we proceed to the refinement stage, where we first render the view using 3D GS, and then predict the relative pose between the target and rendered images to perform pose optimization.

The advantages of the proposed pipeline are twofold. First, we use 3D Gaussians to represent the scene in the initial pose estimation stage. Compared to the point cloud representation in SCR methods, 3D Gaussians retain more detailed geometric and texture information. Additionally, by establishing correspondences between image pixels and 3D Gaussians, GS-RelocNet enables scene-independent relocalization. Second, we employ the rendered view generated by 3D GS for pose refinement, which reduces the domain gap between the rendered and real views. Through this refinement process, the localization accuracy is further enhanced.

## 3.2 Initial Pose Estimation by GS-RelocNet

**Pose estimation process**. In the initial stage, we regress the confidence matrix between image pixels and selected Gaussians by GS-RelocNet. Based on the confidence threshold, we establish the correspondences between pixels and Gausaians, followed by a PnP method with RANSAC to solve the initial pose.

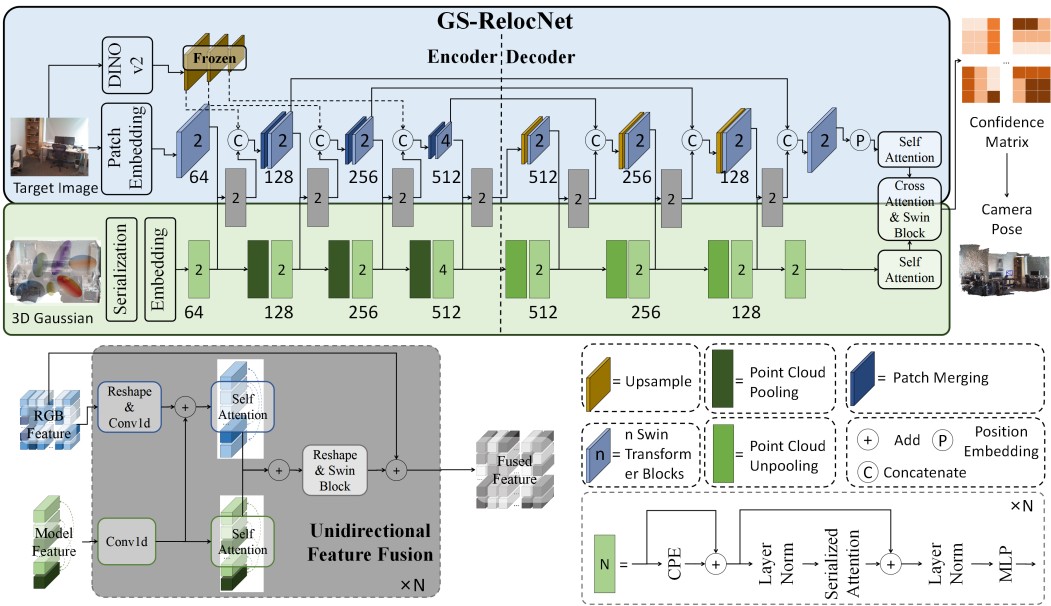

Figure 2: Architecture of GS-RelocNet. The GS-RelocNet framework integrates an RGB feature encoder, a 3D Gaussian feature encoder, an RGB descriptor decoder, a point descriptor decoder, and a confidence metric regression module. In the figure, arrows indicate the process flow, and numbers adjacent to each block denote the corresponding filter size.

**Inputs and outputs of GS-RelocNet**. The GS-RelocNet receives a monocular RGB image and a 3D model expressed by 3D Gaussians as inputs, and outputs the confidence scores that quantify the correspondence between pixels and 3D Gaussians. These confidence scores are then used to establish 2D-3D correspondences for camera pose prediction.

**Input processing**. For 2D image processing, GS-RelocNet employs patch partition and embedding to segment the images into multiple parts. For the 3D Gaussians, the point cloud serialization and embedding are exploited to transform unstructured 3D Gaussians into a structured format. Specifically, the position, alpha, covariance matrix and spherical harmonic function of 3D Gaussian are processed independently, with the features from all four components concatenated. Additionally, we incorporate features from DINO V2 Oquab et al. (2023), along with a depth estimation head, as supplementary input to enhance geometry feature learning. Notably, the parameters of the DINO V2 model are kept frozen during this process.

**Architecture of GS-RelocNet**. The detailed architecture of GS-RelocNet is presented in Fig. 2. It consists of a spatial image encoder, a 3D Gaussian encoder, an image descriptor decoder, a 3D Gaussian descriptor decoder, and a confidence matrix decoder. Within both encoders and decoders, we incorporate a unidirectional feature fusion module to facilitate effective feature sharing from the model to the RGB image. The image branch employs a Swin Transformer Liu et al. (2021) architecture, consisting of multiple Swin Blocks, while the point cloud branch utilizes Point Transformer V3 Wu et al. (2024) for 3D Gaussian feature learning.

**Unidirectional feature fusion module**. Between consecutive image and 3D Gaussian learning blocks, we propose a unidirectional feature fusion module to combine 2D and 3D features, shown at the lower left of Fig. 2. The module takes image features and geometry features as inputs, and outputs the fused features of the two parts. Let the input RGB feature have dimensions $H_u \times W_u \times D_u$, and the model feature have dimensions $N_m \times D_m$. The whole fusion process is as follows.

Step 1, alignment of features. The module aligns the image and 3D model features. Specifically, the RGB feature is reshaped to $N_m \times (H_u \times W_u \times D_u/N_m)$, followed by a 1D convolution to transform it to $N_m \times D_m$, aligning with the model feature dimensions.

Step 2, fusion with self-attention. In this step, the features from the image and 3D Gaussian are first added. Subsequently, multi-head self-attention is applied to added features and original model features respectively. Finally, the both features are again added to achieve feature fusion.

Step 3, feature transformation. The combined feature is reshaped to $H_u \times W_u \times (N_m \times D_m/H_u/W_u)$. A Swin Transformer block is then applied to restore the feature dimensions to $H_u \times W_u \times D_u$, matching the input RGB feature. The input RGB feature is added to this output to produce the merged feature.

Step 4, iterative fusion. Steps 1 through 3 are iteratively applied. Specifically, the fused features from the current iteration are used as the RGB input for the next iteration.

Through these steps, GS-RelocNet can perform one-way fusion of 2D image and 3D Gaussian features. Specifically, the model features can influence the image descriptor learning, but not the other way around. On one hand, this feature sharing enhances RGB feature descriptor learning. On the other hand, the independence of model descriptor learning allows for pre-prediction of model descriptors before the inference stage, significantly accelerating the overall process. In summary, we argue that this fusion mechanism contributes to the localization task, and its effectiveness will be validated in the subsequent experiments.

**Confidence matrix regression**. After regressing the $N_i$ image descriptors and $N_g$ 3D Gaussian descriptors, GS-RelocNet performs regression of a confidence matrix with $N_i * N_g$ scores, where each score represents the confidence between its corresponding image and 3D Gaussian. The regression process proceeds as follows. First, for the image features, GS-RelocNet applies a positional encoding operation followed by a self-attention operation. Similarly, for the 3D Gaussian features, GS-RelocNet also applies a self-attention operation. Next, cross-attention is applied to the two sets of features. Finally, a dual softmax operation is employed to predict the final confidence matrix.

**Loss of GS-RelocNet**. To train GS-RelocNet, we need to construct the ground truth confidence between each 3D Gaussian and image pixel. Given a 3D Gaussian with 2D covariance matrix $\Sigma$ under the current view, then the ground truth confidence is calculated as the following formula,

$$C_g = \frac{1}{2\pi \mid \Sigma \mid} exp[-\frac{1}{2}(x - \mu)^T \Sigma^{-1}(x - \mu)], \tag{1}$$

where $\mu$ denotes the 2D Gaussian center, and $x$ is the pixel position.

## 4 Pose Refinement

**Refinement process**. After processing with GS-RelocNet, we obtain the initial camera pose of the target image. Subsequently, we rendered the current view using 3D GS with the predicted camera pose. The refinement process is to predict the relative pose $(R_r^p, T_r^p)$ between the input real image and rendered view. Finally, the refined pose $(R_f^p, T_f^p)$ is obtained by the following formula.

$$R_f^p = R_r^p * R_i^p, \quad T_f^p = R_r^p * T_i^p + T_r^p. \tag{2}$$

To predict $R_r^p, T_r^p$, we propose a refinement network to regress residual coordinate map proposed in the work Wang & Qi (2023a) between the rendered and real views.

**Refinement network**. As illustrated in Fig. 3, the network takes real and rendered views as inputs, and outputs the residual coordinate map, followed by a PnP method with RANSAC to predict the relative camera pose $(R_r^p, T_r^p)$. The architecture of the refinement network consists of several Swin blocks to learn features of both real and rendered images.

**Bidirectional feature fusion**. Similar to GS-RelocNet, we design a feature fusion module to combine the features of real and rendered images. The key distinction is that the feature fusion process is bidirectional, following these steps. First, the two sets of features are added together, and a self-attention operation is applied to fuse the real and rendered features. Second, a Swin block is used to further process and learn the fused features. Finally, the resulting features of both parts are obtained by adding the original features to the fused output. Notably, the three steps outlined above can be repeated multiple times to refine the feature fusion process.

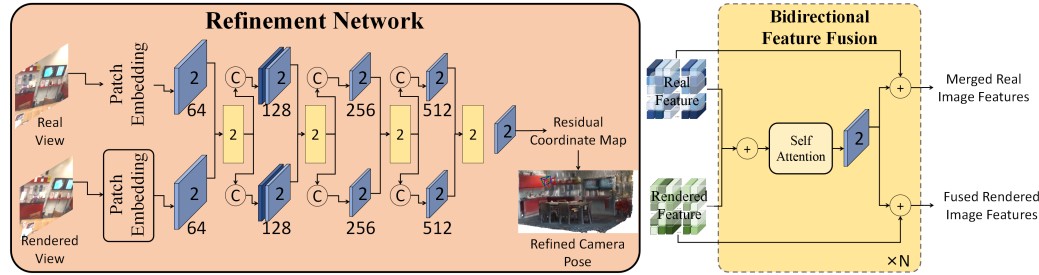

Figure 3: Architecture of the refinement network. The network takes real and rendered views as inputs, and outputs the residual coordinate map to predict the relative camera pose, which is composed of a real image feature encoder, a rendered image feature encoder and a bidirectional fusion module.

**Loss function of refinement network**. For the loss of the residual coordinate ($Loss_{map}$). we use $L_1$ loss to train the refinement network. In addition to the coordinate map loss, the refinement network also exploits the auxiliary loss to facilitate feature learning. Specifically, the pose loss is expressed using the following formula,

$$Loss_{aux} = exp(-T) \cdot \|\mathbf{T}^p - \mathbf{T}^g\| + T + exp(-Q) \cdot \|\mathbf{Q}^p - \mathbf{Q}^g\| + Q, \qquad (3)$$

where $\mathbf{T}^p, \mathbf{T}^g$ represent the prediction and ground truth of camera position, $\mathbf{Q}^p, \mathbf{Q}^g$ mean the prediction and ground truth of camera orientation, and $T, Q$ are variables learned by the refinement network to balance the three terms.

Although the auxiliary loss directly outputs the 6-DoF relative camera pose, it is not used as the final result in our framework. This decision is based on the observation that direct learning based methods typically yield less accurate localization results Wang et al. (2020). Hence the refinement network uses the map to predict the camera pose, with the auxiliary loss as an additional guidance.

$$Loss = Loss_{map} + \alpha * Loss_{aux}. \qquad (4)$$

### 4.1 Datasets and Implementation Details

**Datasets and train-test split**. We conduct experiments on indoor **7 Scenes** Shotton et al. (2013), **TUM RGB-D** Sturm et al. (2012), **Bonn** Palazzolo et al. (2019), **ScanNet** Dai et al. (2017), outdoor **MegaDepth** Li & Snavely (2018) and **Cambridge Landmarks** Kendall et al. (2015). The implementation is divided into scene-dependent and scene-independent settings. In scene-independent setting, we train GS-RelocNet on ScanNet and test it on 7 Scenes, TUM RGB-D and Bonn. For outdoor scene-independent setting, GS-RelocNet is trained on MegaDepth Li & Snavely (2018) and tested on Cambridge Landmarks. When performing the relocalization task on in a scene-dependent manner, GS-RelocNet is trained on 7 Scenes and Cambridge Landmarks respectively.

**Gaussian selection**. In training and inference stages, 4096 Gaussians are selected by spatially uniform sampling. Specifically, we partition the 3D space into $S_x \times S_y \times S_z$ grids with each grid resolution of 0.1m. Let $N_t$ denote the total number of Gaussians. For each grid cell containing $N_g$ Gaussians, we randomly sample $N_g * 4096/N_t$ Gaussians. If the number of sampled Gaussians is less than 4096, we randomly duplicate some samples to reach the desired count for training.

**GS-RelocNet details**. The RGB input of our framework is resized to $256 \times 256$ pixels, while 4096 3D Gaussians spatially uniform sampled from all 3D Gaussians, are processed by the Point Transformer network. In Fig. 2, the module setting and output filter size are indicated near the corresponding modules. Additionally, GS-RelocNet leverages an ADAM W optimizer with learning rates $2 \times 10^{-4}$.

**Refinement network training details**. After obtaining the initial poses, we use 3D GS to render the view. Both the real and rendered images are resized to $128 \times 128$, while the output size of the relative structure is $64 \times 64$. In Fig. 3, the module setting and output filter size are also annotated. The loss coefficient $\alpha$ in Eq. (4) is set to $0.3$, the variables $T, Q$ are initially set to $0.0$.

**Inference details**. After training GS-RelocNet, we use the selected Gaussians and the test image to obtain the confidence map. Then we initially apply a fixed threshold (set to 0.7 in our experiments) to eliminate correspondences with similarity scores below this value. Subsequently, for each pixel

associated with multiple Gaussian correspondences, the pixel's coordinate is computed as a weighted average of the selected Gaussians, weighted by their confidence values. Additionally, if the number of correspondences falls below 100, we re-run GS-RelocNet by selecting an alternative set of 4096 Gaussians from the grids that contain Gaussians with confidence higher than 0.7. After determining the pixel and its weighted Gaussian coordinate, we use PnP with RANSAC to solve the initial pose.

Based on the initial pose, we render the view by trained 3D GS. Given the input image and rendered view, we predict the residual coordinate map by the refinement network, followed by a PnP method with RANSAC to solve the relative pose between them. Finally, the refined pose is obtained by Eq. 2.

## 4.2 Static 7 Scenes

Table 1: Median position (cm) and rotation (°) errors on 7 Scenes. The sign ‡ means the result with SfM pseudo ground truth, while others leverage the original KinectFusion ground truth. In each scene, the red and blue marks represent the first and second.

| | Method | Chess | Fire | Heads | Office | Pumpkin | Kitchen | Stairs | Mean |
|---|---|---|---|---|---|---|---|---|---|
| **Scene-dependent** | | | | | | | | | |
| APR | MS-Transformer Shavit et al. (2021) (ICCV 2021) | 11/6.38 | 23/11.5 | 13/13.0 | 18/8.14 | 17/8.42 | 16/8.92 | 29/10.3 | 18/9.51 |
| | DFNet Chen et al. (2022) (ECCV 2022) | 3/1.12 | 6/2.30 | 4/2.29 | 6/1.54 | 7/1.92 | 7/1.74 | 12/2.63 | 6/1.93 |
| | Marepo Chen et al. (2024b) (CVPR 2024) | 2.1/1.24 | 2.3/1.39 | 1.8/2.03 | 2.8/1.26 | 3.5/1.48 | 4.2/1.71 | 5.6/1.67 | 3.2/1.54 |
| | MS-HyperPose Ferens & Keller (2025) (CVPR 2025) | 11/4.34 | 23/9.79 | 13/10.7 | 17/6.05 | 16/5.24 | 17/6.86 | 27/6.00 | 18/7.00 |
| SCR | ACE‡ Brachmann et al. (2023) (CVPR 2023) | 0.55/0.18 | 0.83/0.33 | 0.53/0.33 | 1.0/0.29 | 1.1/0.22 | 0.77/0.21 | 2.89/0.81 | 1.1/0.34 |
| | DeViLoc Giang et al. (2024) (CVPR 2024) | 2/0.78 | 2/0.74 | 1/0.65 | 3/0.82 | 4/1.02 | 3/1.19 | 4/1.12 | 2.7/1.10 |
| | GLACE‡ Wang et al. (2024a) (CVPR 2024) | 0.6/0.18 | 0.9/0.34 | 0.6/0.34 | 1.1/0.29 | 0.9/0.23 | 0.8/0.20 | 3.2/0.93 | 1.2/0.36 |
| NeRF | CROSSFIRE Moreau et al. (2023) (CVPR 2023) | 1/0.4 | 5/1.9 | 3/2.3 | 5/1.6 | 3/0.8 | 2/0.8 | 12/1.9 | 4/1.10 |
| | NeRFMatch‡ Zhou et al. (2024a) (ECCV 2024) | 0.9/0.30 | 1.1/0.40 | 1.5/1.00 | 3/0.80 | 2.2/0.60 | 1.0/0.30 | 10.1/1.70 | 2.8/0.70 |
| | PMNet Lin et al. (2024a) (ECCV 2024) | 4/1.70 | 10/4.51 | 7/4.23 | 7/1.96 | 14/3.33 | 14/3.36 | 16/3.62 | 10/3.24 |
| 3D GS | DFNet + GS-CPR‡ Liu et al. (2025) (ICLR 2025) | 0.7/0.20 | 0.9/0.32 | 0.6/0.36 | 1.2/0.32 | 1.3/0.31 | 0.9/0.25 | 2.2/0.61 | 1.1/0.34 |
| | ACE + GS-CPR‡ Liu et al. (2025) (ICLR 2025) | 0.5/**0.15** | 0.6/0.25 | 0.4/**0.28** | 0.9/**0.26** | 1.0/**0.23** | 0.7/**0.17** | 1.4/0.42 | 0.8/0.25 |
| | STDLoc‡ Huang et al. (2025) (CVPR 2025) | 0.46/**0.15** | **0.57**/**0.24** | 0.45/**0.26** | **0.86**/**0.24** | **0.93**/**0.21** | 0.63/0.19 | 1.42/**0.41** | 0.76/**0.24** |
| | **Ours‡ (No Refinement)** | **0.44**/0.17 | 0.61/**0.24** | **0.39**/0.30 | 0.89/**0.24** | 0.95/0.28 | **0.60**/0.22 | **1.36**/**0.39** | **0.75**/0.26 |
| | **Ours‡** | **0.41**/**0.15** | **0.55**/**0.21** | **0.37**/**0.26** | **0.85**/**0.24** | **0.92**/0.25 | **0.58**/**0.18** | **1.30**/**0.35** | **0.71**/**0.23** |
| **Scene-independent** | | | | | | | | | |
| Hand-crafted | Active Search Sattler et al. (2016) (TPAMI) | 4/1.96 | 3/1.53 | 2/1.45 | 9/3.61 | 8/3.10 | 7/3.37 | 3/2.22 | 51/2.46 |
| RPR | RelocNet Balntas et al. (2018) (ECCV 2018) | 21/10.90 | 32/11.80 | 15/13.40 | 31/10.30 | 40/10.90 | 33/10.30 | 33/11.40 | 29.3/11.29 |
| | Relative PoseNet Laskar et al. (2017) (ICCV 2017) | 31/15.00 | 40/19.00 | 24/22.20 | 38/14.10 | 44/18.20 | 41/16.50 | 35/23.60 | 36.1/18.37 |
| SCR | InLoc Taira et al. (2018) (CVPR 2018) | 3/1.05 | 3/1.07 | 2/1.16 | 3/1.05 | 5/1.55 | 4/1.31 | 9/2.47 | 4.1/1.38 |
| | Pixloc Sarlin et al. (2021) (CVPR 2021) | 2/**0.80** | 2/0.73 | 1/0.82 | 3/**0.82** | 4/1.21 | 3/1.20 | 5/1.30 | 2.9/0.98 |
| | Wang et al. Wang & Qi (2023a) (ISMAR 2023) | 2.4/0.97 | 2.0/0.99 | 1.6/1.27 | 2.4/1.01 | 3.7/1.20 | 2.8/1.14 | 3.1/1.22 | 2.6/1.11 |
| | DUSt3R Wang et al. (2024b) (CVPR 2024) | 3/0.96 | 4/1.02 | 1/1.00 | 4/1.04 | 5/1.26 | 4/1.36 | 21/4.06 | 6/1.53 |
| | Reloc3R Dong et al. (2025) (CVPR 2025) | 3/0.99 | 4/1.13 | 2/1.23 | 5/**0.88** | 7/1.14 | 5/1.23 | 12/1.25 | 5.4/1.12 |
| 3D GS | **Ours (No Refinement)** | **1.3**/0.81 | **1.2**/0.65 | **0.7**/0.73 | **1.6**/0.89 | **2.7**/1.01 | **2.5**/1.10 | **2.1**/**1.00** | **1.7**/0.88 |
| | **Ours** | **1.0**/0.72 | **1.0/0.64** | **0.6/0.70** | **1.4/0.82** | **2.2/0.93** | **2.0/1.02** | **1.9/0.92** | **1.4/0.82** |

In Table 1, we provide the experimental localization results on 7 Scenes. with both scene-dependent and scene-independent categories using the median position and rotation errors.

**Scene-dependent method comparison**. In scene-independent setting, our results gain best accuracy on both mean position and orientation metrics. Among 7 scenes, our method obtains the best position on all 7 scenes and orientation on 5 scenes, demonstrating the state-of-the-art performance under scene-dependent situation. Additionally, our method achieves $1.21cm/0.61°$ with the original ground truth, which also demonstrates superior accuracy compared to other methods evaluated.

**Scene-independent method comparison**. With scene-independent approaches, our method achieves the lowest mean position and orientation errors on all 7 scenes. Moreover, our method significantly reduces the position error from other best $2.6cm/0.98°$ to $1.4cm/0.82°$ ($\downarrow 46.2\%/16.3\%$). Besides the localization accuracy, to our knowledge, our framework is the first scene-independent relocalization method with 3D GS.

**Result visualization**. Fig. 4 provides visualization results of the camera trajectories with green poses denoting the ground truth and blue ones representing our results. The results show minimal differences between the predicted and ground truth poses, demonstrating the suitability of our method.

## 4.3 Dynamic TUM RGB-D and Bonn

**Challenges on TUM RGB-D and Bonn**. The TUM RGB-D test sequences involve two individuals walking around a table increase the complexity of localization. Similarly, the Bonn dataset features highly dynamic sequences, such as individuals manipulating boxes or interacting with balloons.

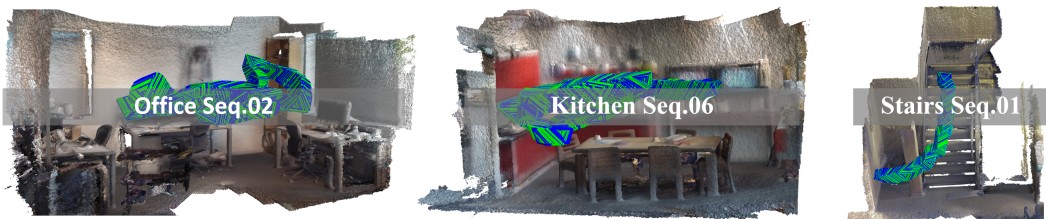

Figure 4: The visualization results of camera pose on 7 Scenes. In each scene, the green and blue poses denote the ground truth and prediction respectively.

Table 2: RMSE of ATE [cm] results in four dynamic scenes of TUM RGB-D. In each scene, the red and blue marks represent the first and second respectively.

| | Method | fr3_walking_ | | | | Mean |
| | | xyz | static | rpy | half | |
|---|---|---|---|---|---|---|
| Hand-crafted | ORB-SLAM2 Mur-Artal & Tardós (2017) | 45.9 | 9.3 | 65.8 | 32.8 | 38.5 |
| | DGM-VINS Song et al. (2023) | 3.6 | 1.3 | 7.1 | 3.3 | 3.8 |
| Hand-crafted + Semantics | DynaSLAM Bescos et al. (2018) | **1.5** | **0.6** | **3.5** | **2.5** | **2.0** |
| | DS-SLAM Yu et al. (2018) | 2.5 | 0.8 | 44.4 | 3.0 | 12.7 |
| | LC-CRF SLAM Du et al. (2020) | 1.6 | 1.1 | 4.6 | 2.8 | 2.5 |
| 3D GS | Ours | **1.1** | **0.4** | **2.2** | **2.0** | **1.4** |

**Root Mean Square Error (RMSE) of Absolute Trajectory Error (ATE) on dynamic TUM RGB-D**. Table 2 presents the RMSE of ATE for four dynamic scenes, compared against ORB-SLAM2, DynaSLAM, DS-SLAM, LC-CRF SLAM, and DGM-VINS. Our approach consistently outperforms these state-of-the-art SLAM systems in the RMSE of ATE metric, demonstrating superior localization accuracy in dynamic environments.

**RMSE of ATE on dynamic Bonn**. We evaluated our framework on the Bonn dataset across 20 test scenes, consistent with LC-CRF SLAM, and compared it with ReFusion Palazzolo et al. (2019), MaskFusion Runz et al. (2018), and LC-CRF SLAM Du et al. (2020). The mean RMSE of ATE results are 23.8cm (ReFusion), 25.1cm (MaskFusion), 6.8cm (LC-CRF SLAM), and 4.3cm (Ours). Our framework achieves the lowest mean RMSE of ATE, highlighting its exceptional localization accuracy in highly dynamic scenes.

## 4.4 Cambridge Landmarks

Table 3: Median localization results on Cambridge Landmarks compared with other methods. Units of position and orientation are centimeter (cm) and °. ⋆ means the result with a scene-independent setting. In each scene, the red and blue marks represent the first and second.

| | Method | College | Hospital | Shop | StMary | Mean |
|---|---|---|---|---|---|---|
| APR | MS-Transformer Shavit et al. (2021) (ICCV 2021) | 85/1.45 | 175/2.43 | 88/3.20 | 166/4.12 | 129/2.80 |
| | DFNet Chen et al. (2022) (ECCV 2022) | 73/2.37 | 200/2.98 | 67/2.21 | 137/4.02 | 119/2.90 |
| SCR | InLoc Taira et al. (2018) (CVPR 2018) | 46/0.8 | 48/1.0 | 11/0.5 | 18/0.6 | 31/0.73 |
| | DSAC* Brachmann & Rother (2021) (TPAMI) | 15/0.3 | 21/0.4 | 5/0.3 | 13/0.4 | 14/0.35 |
| | ACE Brachmann et al. (2023) (CVPR 2023) | 29/0.38 | 31/0.61 | 5/0.3 | 19/0.6 | 21/0.47 |
| | DUSt3R-224⋆ Wang et al. (2024b) (CVPR 2024) | 20/0.32 | 26/0.46 | 9/0.38 | 11/0.38 | 17/0.39 |
| | Reloc3R-224⋆ Dong et al. (2025) (CVPR 2025) | 47/0.41 | 87/0.66 | 18/0.53 | 41/0.73 | 48/0.58 |
| NeRF | NeuMap Tang et al. (2023) (CVPR 2023) | 14/0.2 | 19/0.4 | 6/0.3 | 17/0.5 | 14/0.35 |
| | CROSSFIRE Moreau et al. (2023) (ICCV 2023) | 47/0.7 | 43/0.7 | 20/1.2 | 39/1.4 | 37/1.00 |
| | NeRFMatch Zhou et al. (2024b) (ECCV 2024) | 12.7/0.2 | 20.7/0.4 | 8.7/0.4 | 11.3/0.4 | 0.13/0.35 |
| | PMNet Lin et al. (2024a) (ECCV 2024) | 68/1.97 | 103/1.31 | 58/2.10 | 133/3.73 | 91/2.28 |
| 3D Gaussian | DFNet + GS-CPR Liu et al. (2025) (ICLR 2025) | 26/0.34 | 48/0.72 | 10/0.36 | 27/0.62 | 28/0.51 |
| | ACE + GS-CPR Liu et al. (2025) (ICLR 2025) | 25/0.29 | 26/0.38 | 5/0.23 | 13/0.41 | 17/33 |
| | STDLoc Huang et al. (2025) (CVPR 2025) | 15/**0.17** | **11.9/0.21** | 3/**0.13** | **4.7/0.14** | **9/0.16** |
| | Ours (No Refinement) | **11**/0.19 | 13/0.26 | **4**/0.18 | 7/0.15 | **9/0.20** |
| | Ours⋆ | 12/0.18 | 13/0.25 | 5/0.19 | 7/0.20 | **9**/0.21 |
| | Ours | **9/0.15** | **10/0.19** | **3/0.15** | **5/0.13** | **7/0.16** |

In Table 3, we provide the experimental localization results on Cambridge Landmarks in comparison with APR, SCR, NeRF and 3D GS methods. The results on Cambridge Landmarks demonstrate that our method gains the best performance on both mean position and orientation metrics. Compared

with the recent PMNet Lin et al. (2024a), the position error is significantly reduced from 91cm to 7cm, further validating the effectiveness of our framework.

**Generalization on Cambridge Landmarks**. To assess generalization, we trained GS-RelocNet on the MegaDepth dataset Li & Snavely (2018) and tested it on the Cambridge Landmarks dataset in a scene-independent setting (marked as $^\dagger$). The mean pose error across four scenes is 9cm/0.21°, surpassing the performance of DUSt3R (17cm/0.39°) and Reloc3R (48cm/0.58°). These results underscore the robustness and generalization capability of GS-RelocNet in diverse, unseen environments.

**Discussion of pose refinement**. In Tables 1 and 3, we also present the localization performance without pose refinement. Two notable observations emerge from the results. First, pose refinement leads to improvements in localization accuracy across all scenes, demonstrating the positive impact of the refinement process using 3D Gaussians. Second, even without pose refinement, our method remains competitive with other state-of-the-art approaches. On 7 Scenes, the scene-dependent result is comparable with STDLoc, while the scene-independent performance is obviously more accurate than others. On Cambridge, the results are also comparable to those of STDLoc.

**Discussion of running time**. Our framework efficiency comprises initial pose estimation with confidence map regression and PnP, and pose refinement with view rendering, residual map regression and PnP. On average, it processes testing images at 65 ms (15.4 FPS) on an Nvidia 4090 GPU across 7 Scenes and Cambridge Landmarks. The average running times per frame are as follows, 39 ms for confidence regression, 4 ms for initial PnP with RANSAC, 9 ms for view rendering, 8 ms for residual coordinate regression, and 5 ms for PnP in pose refinement. This outperforms 3D GS based methods like ACE Brachmann et al. (2023) + GS-CPR Liu et al. (2025) (190ms, 5.3 FPS) and STDLoc Huang et al. (2025) (143ms, 7 FPS), highlighting our computational efficiency.

**Discussion of comparison with 3G GS based methods**. From Tables 1 and 3, we can see that localization performance of 3D GS based STDLoc and ACE + GS-CPR is slightly less accurate than ours, but also is competitive. Compared to these methods, the additional superiorities of our framework lies in two aspects. First, our method supports scene-independent relocalization, while STDLoc and GS-CPR requests training before localization in the target scene. Second, our inference speed is faster than the other two methods, achieving more than twice speed.

## 4.5 Detailed Studies

Table 4: Scene-independent localization results on 7 Scenes with original ground truth and Cambridge Landmarks with different settings.

| | Fusion in encoder | Fusion in decoder | Fusion in Refinement | 7 Scenes | Cambridge Landmarks |
|---|---|---|---|---|---|
| S1 | × | × | × | 2.1/1.17 | 14/0.35 |
| S2 | √ | × | × | 1.9/1.14 | 13/0.32 |
| S3 | × | √ | × | 1.9/1.11 | 12/0.36 |
| S4 | × | × | √ | 2.0/1.03 | 13/0.35 |
| S5 | × | √ | √ | 1.6/0.97 | 11/0.28 |
| S6 | √ | × | √ | 1.7/0.92 | 10/0.26 |
| S7 | √ | √ | × | 1.5/0.84 | 9/0.23 |
| S8 | √ | √ | √ | 1.4/0.82 | 9/0.21 |
| | Iterations of fusion model in GS-RelocNet | | | | |
| S9 | 1 | | | 1.7/0.92 | 12/0.29 |
| S10 | 2 | | | 1.4/0.82 | 9/0.21 |
| S11 | 4 | | | 1.4/0.86 | 10/0.22 |
| S12 | 8 | | | 1.5/0.87 | 11/0.24 |

**Discussion of outstanding performance**. The results of the aforementioned experiments demonstrate state-of-the-art performance of our framework. We attribute our outstanding performance to three main factors. First, our framework leverages the full potential of 3D GS for both initial pose estimation and pose refinement. Second, GS-RelocNet is specifically designed to establish accurate correspondences between pixels and 3D Gaussians. Third, we propose a refinement network that predicts the relative pose between real and rendered images using a bidirectional feature fusion module. In the following sections, we present ablation experiments to demonstrate the effectiveness of these fusion modules.

**Ablation studies of fusion modules**. In S1 - S8 of Table 4, we conduct ablation experiments of the three fusion modules, including the unidirectional feature fusion module in encoder, decoder of

GS-RelocNet and the bidirectional fusion module in the refinement network. In S5, S6, and S7, each module is removed individually. With this setting, the two features are reshaped and concatenated directly. It is evident that accuracy across all three datasets decreases in comparison to setting S8. In S2, S3, and S4, two modules are removed individually, and the results are less accurate than those in settings S5, S6, and S7. In setting S1, where all three modules are removed, the accuracy decreases most significantly, further confirming the positive impact of the three fusion modules.

**Detailed studies of the iterations in the unidirectional feature fusion module**. In S9 - S12 of Table 4, we explore the effect of the number of iterations in the unidirectional feature fusion module by setting the iteration values to 1, 2, 4, and 8, respectively. When the iteration is set to 1, the result demonstrates that the estimation performance is less accurate than the other 3 settings. Moreover, the results for iterations set to 2, 4, and 8 are comparable. This discrepancy can be explained by the fact that with only one iteration, the RGB and point cloud features are not sufficiently fused, resulting in less accurate estimates. In contrast, with 2, 4, or 8 iterations, the fusion between the two feature types is sufficiently accomplished for RGB based pose estimation, leading to comparable performance.

**Limitation discussion**. A primary limitation of our framework is its reliance on a high-quality 3D GS model of the target scene. When the 3D GS model is of suboptimal quality, localization performance may degrade, leading to failures or significant errors.

## 4.6   AR Application

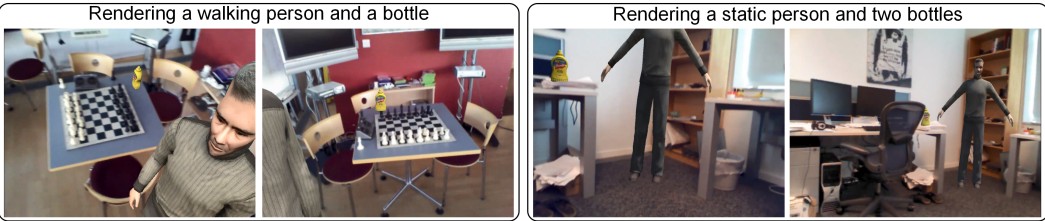

Figure 5: AR effect on the Chess and Office scene of 7 Scenes. We render a virtual walking person and a bottle onto the Chess scene, and two virtual bottles and a static person onto the Office scene based on the predicted camera pose of our framework.

To demonstrate the performance in real-world AR applications, we present the virtual-real fusion results for two scenes using the predicted pose by our framework, the Chess and Office scenes from the 7 Scenes dataset, as shown in Fig. 5. Specifically, we render a virtual walking person and a bottle onto the Chess scene, and two virtual bottles and a static person onto the Office scene.

## 5   Conclusion

In this paper, we propose a novel 3D Gaussian based camera relocalization framework, composed of of two stages, an initial pose estimation stage, which predicts 2D-3D correspondences between image pixels and 3D Gaussians, and a refinement stage, which estimates the residual pose between the target and rendered views. To estimate the 2D-3D correspondences, we introduce a descriptor matching network called GS-RelocNet. Within GS-RelocNet, we design a unidirectional feature fusion model to combine RGB features with Gaussian features. After obtaining the initial camera pose, we proceed with the pose refinement network. In this refinement network, we propose a bidirectional feature fusion model to merge the features from the real and rendered images. To validate the performance of our framework, we conduct experiments on both indoor 7 Scenes, TUM RGB-D, Bonn and outdoor Cambridge Landmarks datasets. The results demonstrate state-of-the-art localization accuracy on both datasets. Additionally, we provide detailed studies on the feature fusion modules and the refinement stage, further highlighting the effectiveness of our approach.

In summary, this paper makes a significant contribution by introducing a scene-independent localization framework through the full utilization of 3D Gaussians. By leveraging 3D Gaussians in both the initial and refinement stages, our method is able to deliver more accurate localization results across a variety of scenes. We hope that the proposed framework could be a universal localization pipeline.

# Acknowledgments

The paper is supported by Shandong Provincial Natural Science Foundation (No. ZR2024QF215), Key Research and Development Program of Rizhao (No. 2024ZDYF010053), National Natural Science Foundation of China (No. 62072020) and the Open Project Program of State Key Laboratory of Virtual Reality Technology and Systems, Beihang University (No. VRLAB2024A**).

The authors thank the Zhiyang Innovation Technology Co., Ltd. for computing power and data support.

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

# A Technical Appendices and Supplementary Material

## A.1 Residual Coordinate Map for Pose Refinement

Given a pair of images, the residual coordinate map denotes the XYZ coordinate difference between the current and previous camera coordinate space Wang & Qi (2023a), which is used to predict the relative pose. In our paper, given the real input image and rendered view with initially predicted pose $R_i^p, T_i^p$, we use the residual coordinate map to solve the relative pose $R_r^p, T_r^p$ between them. Then the refined pose is obtained by Eq. 2.

The construction process of residual coordinate map is as follows. We substitute the depth information (Z-axis value) with the grayscale value $M_d$ denotes the XYZ coordinates under the camera space of the input real view, obtained by uniformly sampling from the grayscale image. For each point $\mathbf{p} \in M_d$, we first transform it to the world space. Then, it is converted to the camera space of the rendered frame. Finally, the relative point is obtained by subtracting the original coordinate from the transformed one. In summary, the coordinate representation is defined by the following formula.

$$M_r = (R_r^p - E)M_d + T_r^p, \tag{5}$$

where $E$ denotes the identity matrix. Through regressing the coordinate map by the refinement network, $R_r^p, T_r^p$ can be predicted by the PnP method with RANSAC.

## A.2 More Dataset Details

**7 Scenes** Shotton et al. (2013). This dataset includes seven indoor scenes, each containing 2 to 10 sequences. It provides depth images, color frames, and ground truth poses. All scenes were recorded using a handheld Kinect RGB-D camera at a resolution of $640 \times 480$ and are divided into separate training and testing sets. The ground truth poses were obtained using the KinectFusion system. This dataset has recently been used as a benchmark in studies Kendall & Cipolla (2017); Brachmann & Rother (2018); Zhou et al. (2020), making it convenient for comparison with other methods.

**Cambridge Landmarks Kendall et al. (2015).** This outdoor dataset includes several large outdoor environments. In this paper, we use five scenes (College, Hospital, Shop, and Church) to evaluate localization accuracy. Ground truth poses in each scene were calibrated using Visual SFM, and a sparse point cloud is also provided.

**ScanNet** Dai et al. (2017). ScanNet contains over 1,500 scans, amounting to around 2.5 million views. The dataset was captured using a user-friendly and scalable RGB-D capture system. To evaluate scene-independent performance, we train GS-RelocNet on ScanNet and tested it on the 7 Scenes dataset.

**TUM RGB-D** Sturm et al. (2012). The TUM RGB-D dataset is designed to benchmark visual odometry and SLAM systems. We select the four dynamic scenes (fr_walking_xyz, fr_walking_static, fr_walking_rpy, fr_walking_half) from TUM RGB-D to evaluate the localization performance in dynamic environments.

**Bonn** Palazzolo et al. (2019). The Bonn dataset is tailored for dynamic localization, featuring highly dynamic sequences. We select 20 sequences from the dynamic subset (same as LC-CRF SLAM Du et al. (2020)), where individuals perform various tasks such as manipulating boxes or interacting with balloons, along with 2 static sequences.

**12 Scenes** Valentin et al. (2016). The 12 Scenes dataset features 12 larger indoor environments, with volumes ranging from $14m^3$ to $79m^3$.

## A.3 More Implementation Details

**3D GS training details**. To construct the 3D GS model, we first utilize COLMAP to generate an initial point cloud using ground truth poses. Subsequently, we employ the original 3D GS model with its default configuration settings.

–iterations: 30000. Total number of training iterations.

–position_lr_init: 0.00016. The initial learning rate of the Gaussian position.

–position_lr_final: 0.0000016. The final learning rate of the Gaussian position.

–position_lr_delay_mult: 0.01. The delay multiplier before the learning rate decay begins.

–position_lr_max_steps: 30000. The total number of steps for the learning rate decay.

–feature_lr: 0.0025. Learning rate of spherical harmonic function coefficients.

–opacity_lr: 0.05. Learning rate of opacity.

–scaling_lr: 0.005. Scaling learning rate.

–rotation_lr: 0.001. Learning rate of rotation.

–densify_from_iter: 500. Densification begins from which iteration.

–densify_until_iter: 15000. Densification ends at which iteration.

–densification_interval: 100. Perform densification and pruning checks every few iterations.

–opacity_prune_threshold: 0.005. Opacity pruning threshold.

–densify_grad_threshold: 0.0002. The gradient threshold for densifying the Gaussian sphere.

**PnP details**. The PnP with RANSAC uses OpenCV implementation with following parameters.

–iterationsCount: 100. The number of RANSAC iterations.

–reprojectionError: 8. Threshold for reprojection error.

–confidence: 0.99. Degree of confidence.

–flags:SOLVEPNP_ITERATIVE. PnP solver algorithm.

**Pose extension in the refinement network**. In the refinement network training, small pose differences between rendered and real frames require high coordinate accuracy, which increases the learning difficulty. To address this, we extend the relative pose using fixed coefficients. The position is scaled directly, and the orientation is expanded through a transformation between the quaternion and Euler angles. Specifically, the extension coefficient is set to 8.0.

**Initial pose estimation by PnP**. For the PnP solver with RANSAC, we adapt the traditional RANSAC framework by incorporating our predicted confidence scores. Conventionally, RANSAC determines the final result based on the number of inlier points. In our modified approach, we instead use the sum of the confidence values of the inlier points to make this determination, thereby improving the reliability of the pose estimation.

### A.4 More Results on 7 Scenes

Table 5: The percentage of localization error under $5cm, 5°$ and $2cm, 2°$ on indoor 7 Scenes compared with other methods. Specially, the sign $\ddagger$ means the result with SfM pseudo ground truth, while others leverage the original KinectFusion ground truth. The red and blue marks represent the first and second.

| | Method | $5cm, 5°$ (↑) | $2cm, 2°$ (↑) |
|---|---|---|---|
| APR | DFNet Chen et al. (2022) | 43.1 | 8.4 |
| | Marepo Chen et al. (2024b) | 84.0 | 33.7 |
| SCR | DSAC*$^\ddagger$ Brachmann & Rother (2021) | 97.8 | 80.7 |
| | ACE$^\ddagger$ Brachmann et al. (2023) | 97.1 | 83.3 |
| | GLACE$^\ddagger$ Wang et al. (2024a) | 95.6 | 82.2 |
| NeRF | NeReS Chen et al. (2024a) | 78.3 | 45.9 |
| | HR-APR Liu et al. (2024b) | 76.4 | 40.2 |
| | NeRFMatch$^\ddagger$ Zhou et al. (2024b) | 78.4 | - |
| 3D Gaussian | DFNet + GS-CPR$^\ddagger$ Liu et al. (2025) (Accepted by ICLR 2025) | 94.2 | 76.5 |
| | ACE + GS-CPR$^\ddagger$ Liu et al. (2025) (Accepted by ICLR 2025) | **100.0** | **93.1** |
| | STDLoc$^\ddagger$ Huang et al. (2025) (Accepted by CVPR 2025) | 99.1 | 90.9 |
| | DFNet + GS-CPR$^\ddagger$ Liu et al. (2024a) | 94.2 | 76.5 |
| | Ours$^\ddagger$ | **99.8** | **94.9** |

**5cm, 5° and 2cm, 2° metric on 7 Scenes**. Besides the accuracy, the localization stability is also an important metric, always expressed by the percentage of position and orientation error under $5cm, 5°$ and $2cm, 2°$. Table 5 presents a comparison of our results with those of DFNet Chen et al. (2022),

Marepo Chen et al. (2024b), DSAC* Brachmann & Rother (2021), ACE Brachmann et al. (2023), GLACE Wang et al. (2024a), NeReS Chen et al. (2024a), HR-APR Liu et al. (2024b), NeRFMatch Zhou et al. (2024b) and GSLoc Liu et al. (2024a).

Our approach achieves an accuracy of 99.8%, slightly below the state-of-the-art performance of ACE Brachmann et al. (2023) + GS-CPR Liu et al. (2025) (100%). However, our method outperforms all other competing approaches. For the more stringent $2cm, 2°$ metric, our framework demonstrates at least a 1.7% improvement in accuracy compared to the next-best method, underscoring its robustness in challenging indoor environments. In comparison to DFNet Chen et al. (2022) + GS-CPR Liu et al. (2025), our method consistently achieves higher accuracy across both metrics. Notably, while GS-CPR relies on accurate initial pose estimates, our approach excels independently, demonstrating superior generalization without requiring such priors.

## A.5   Results on 12 Scenes

Table 6: The percentage of localization error under $2cm, 2°$ on 12 Scenes compared with other methods. The red and blue marks represent the first and second. The results are reported in Liu et al. (2024a).

|  | Method | $2cm, 2°$ (↑) |
|---|---|---|
| APR | Marepo Chen et al. (2024b) | 50.4 |
| SCR | DSAC* Brachmann & Rother (2021) | 96.7 |
|  | ACE Brachmann et al. (2023) | 97.2 |
|  | GLACE Wang et al. (2024a) | **97.5** |
| 3D Gaussian | Marepo Chen et al. (2024b) + GS-CPR Liu et al. (2025) | 90.9 |
|  | Ours | **98.7** |

$2cm, 2°$ **metric on 12 Scenes**. To further evaluate localization performance, we conduct experiments on the 12 Scenes dataset using the $2cm, 2°$ metric. Table 6 presents the results in comparison with other approaches. Our method achieves the highest accuracy in localization for the $2cm, 2°$ metric. Compared to the 3D Gaussian based refinement method, GS-CPR Liu et al. (2025), our approach demonstrates achieves a 7.6% improvement.

## A.6   More Detailed Studies of GS-RelocNet

Table 7: Scene-independent localization results on 7 Scenes with original ground truth and Cambridge Landmarks with different settings.

|  | GS-RelocNet 3D Gaussian Number | 7 Scenes | Cambridge Landmarks |
|---|---|---|---|
| S1 | 512 | 2.4/1.14 | 14/0.36 |
| S2 | 1024 | 2.0/0.94 | 12/0.29 |
| S3 | 2048 | 1.7/0.89 | 10/0.22 |
| S4 | 4096 | 1.4/0.82 | 9/0.21 |
| S5 | 8192 | 1.5/0.88 | 9/0.21 |
| S6 | 16384 | 1.9/0.97 | 12/0.28 |
| S7 | Using Point Cloud instead of 3D Gaussians | 2.5/1.43 | 16/0.45 |
|  | Refinement Network Rendering Method |  |  |
| S8 | 3D model | 1.8/0.92 | 12/0.28 |
| S9 | NeRF | 1.6/0.84 | 10/0.23 |
| S10 | 3D GS | 1.4/0.82 | 9/0.21 |

**Discussion of 3D Gaussian number in GS-RelocNet**. In S1 - S6 of Table 7, we experiment with different numbers of 3D Gaussians. On one hand, the results with 512, 1024, and 16384 Gaussians are inferior to those with 4096 Gaussians, indicating that fewer Gaussians are insufficient for learning the scene features, while too many Gaussians increase the learning difficulty. On the other hand, the results with 2048, 4096, and 8192 Gaussians are comparable, suggesting that these configurations are sufficient for adequately learning the scene.

Subsequently, a common concern may lie in the large scene with 4096 Gaussians. To address this, our framework may adopt a coarse-to-fine strategy. In the coarse stage, we uniformly sample 3D

Gaussians across the entire scene and select those with high confidence scores, iterating this process as needed. In the refinement stage, we focus on Gaussians in proximity to the high-confidence selections, establishing 2D-3D correspondences between image pixels and these 3D Gaussians, followed by PnP to estimate the camera pose.

**Discussion of 3D Gaussian sampling strategy**. For the sampling strategy, we uniformly sample 3D Gaussians within grid cells to ensure a sufficient number of correspondences for robust pose estimation. Within each grid cell, Gaussians are randomly selected for training. Although we use 4096 Gaussians per training iteration, multiple iterations allow most Gaussians to be utilized. To evaluate the robustness of this strategy, we conducted experiments using random sampling instead of uniform grid based sampling. The results show mean pose errors of $0.76cm/0.26°$ in the scene-dependent setting on the 7-Scenes dataset and $7cm/0.18°$ on the Cambridge Landmarks dataset. These results indicate minimal performance degradation, demonstrating the robustness of our sampling strategy.

**Discussion of usage of 3D Gaussians**. In S7 of Table 7, we replace 3D Gaussians with the point cloud of the scene. The results clearly show a significant decrease in localization accuracy compared to using 3D Gaussians. We believe this is due to the fact that 3D Gaussians retain more texture and illumination information, while point clouds only capture geometric details.

**Discussion of Spherical Harmonics (SH) in 3D GS**. To evaluate the contribution of SH in 3D GS, we exclude SH and perform experiments on the 7 Scenes and Cambridge Landmarks datasets without using SH as the input of GS-RelocNet. The scene-independent results reveal a consistent increase in localization error. On 7 Scenes, the error rises from $1.4/0.82$ to $1.7/0.93$ ($\uparrow 0.3/0.11$), and on Cambridge Landmarks, it increases from $9/0.21$ to $10/0.26$ ($\uparrow 1/0.05$). This observed increase in error across both datasets confirms the effectiveness of SH in enhancing the performance of 3D GS for localization tasks.

**Discussion of DINO v2 features**. In GS-RelocNet, we also DINO v2 features as an additional input. To validate its effect, we conduct experiments without DINO v2 features. The results are $0.73/0.25$ ($\uparrow 0.02/0.02$ with scene-dependent setting), $1.4/0.84$ ($\uparrow 0.0/0.02$ with scene-independent setting) on 7 Scenes and $10/0.24$ ($\uparrow 1/0.03$) on Cambridge Landmarks with scene-independent setting. We can see that the additional feature plays a slightly positive role on localization improvement. In the scene-dependent setting on 7 Scenes, it is even negligible. As an explanation, this is because that ScanNet contains large samples for training, making GS-RelocNet can learn the additional features itself.

### A.7   More Detailed Studies of Refinement

**Discussion of rendering method**. In S8 - S10 of Table 7, we present results using rendered images from both the 3D model and the NeRF network. The results demonstrate that the 3D GS based refinement yields the most significant improvements. When using rendered images from the 3D model, the results are comparable to those without refinement, likely due to the domain gap between rendered and real views. By incorporating NeRF and 3D GS, the lighting conditions are also considered, reducing this gap.

**Discussion of cross-attention layers**. To explore the potential of cross-attention layers, we conducted two ablation studies in the pose refinement stage. First, we integrated a cross-attention layer into the bidirectional feature fusion module of the refinement network. This yielded mean pose errors of $0.73cm/0.24°$ on the 7 Scenes dataset and $7cm/0.17°$ on the Cambridge Landmarks dataset in the scene-dependent setting, which are comparable to our original results. Second, we replaced the refinement network with Dust3R ViT variant, incorporating cross-attention. The scene-dependent errors are $0.76cm/0.27°$ on 7 Scenes and $9cm/0.21°$ on Cambridge Landmarks, indicating slightly reduced accuracy. We attribute the limited impact of cross-attention layers to the small input resolution of the refinement network ($128 \times 128$), which constrains their effectiveness.

**Discussion of environment conditions**. Our current implementation does not include specific adaptations for environmental changes, such as variations in lighting or weather. Upon reviewing other 3D GS based methods, including GS-CPR and STDLoc, we found no explicit mention of specialized designs addressing such conditions. To mitigate the impact of environmental variations, we propose exploring 3D GS variants designed for enhanced robustness to environmental changes, as demonstrated in recent works Zhang et al. (2024); Kulhanek et al. (2024). These approaches could be integrated into our framework to improve performance under diverse conditions.

**Discussion of RANSAC**. To predict the initial pose, we use the PnP with RANSAC using predicted confidence. To validate this, we also make the experiments with the traditional RANSAC using inlier number. The scene-independent results are $1.6/0.86$ ($\uparrow 0.1/0.04$) and $10/0.24$ ($\uparrow 1/0.03$) on Cambridge Landmarks. The results show a slight accuracy improvement with confidence in RANSAC.

