# OpenReview forum: "3D Gaussian Splatting based Scene-independent Relocalization with Unidirectional and Bidirectional Feature Fusion"
_NeurIPS.cc/2025/Conference — NeurIPS 2025 poster_

### Official Review · Reviewer_54m2 · 2025-06-17

**Clarity:** 3
**Significance:** 3
**Originality:** 3
**Rating:** 4
**Confidence:** 3

**Summary:**

This paper proposes a camera relocalization system coupled with unidirectional and bidirectional feature fusion module on 3DGS scenes. It extends the state of the art in neural rendering and pose estimation by allowing robust relocalization in unseen environments without scene-specific training.
The pipeline is structured in two stages:
1. A coarse pose estimation module, GS-RelocNet, is used to obtain the confidence matrix between image pixels and randomly selected Gaussians. The confidence matrix is further processed with PnP with RANSAC algorithm to obtain the initial camera pose. GS-RelocNet is trained by comparing with the groundtruth confidence matrix.
2. A rendered view under the predicted camera pose is processed together with the groundtruth rendering by a refinement network. This network outputs the residual coordinate map, also followed by a PnP method with RANSAC to predict the relative camera pose. Refined pose is calculated based on the original pose and relative pose. The refinement network is trained with coordinate map loss and an auxiliary loss for camera’s position and orientation.

Performance on existing datasets claim high localization accuracy and efficiency (~15 fps). The refinement network appears to improve significantly compared to scene-independent methods.

**Questions:**

1. The paper samples 3D Gaussian components uniformly within grid cells to construct correspondences. However, different 3D Gaussian components may have different scales and sizes, which in turn may influence their respective projected area on the image plane. This variation could affect the confidence scores within the confidence matrix and subsequently the final pose estimation. How sensitive is your method’s performance to the sampling strategy of 3D Gaussian components with different sizes?

2. The pipeline utilizes rendered views from 3D Gaussian. However, rendered images may be sensitive to environment conditions, (e.g. lighting directions or intensities, artifacts from the original 3DGS scene). How robust is your method to these variations in rendered feature maps?

3. This pipeline is trained on limited datasets. Under what conditions does the system fail (e.g., low-texture regions, reflective surfaces)?

**Ethical Concerns:**

["NO or VERY MINOR ethics concerns only"]

**Final Justification:**

The authors added further analysis and experimental result to address my concerns. Including these in the final version would enhance the quality of the submission.

**Limitations:**

The paper assumes the scene is static. The method’s ability to deal with moving objects, structural modifications, or other dynamic components — a common scenario in real-world applications — is currently unclear or unaddressed.

**Quality:**

3

**Strengths And Weaknesses:**

Strengths:
1. Scene-independent relocalization for 3DGS scenes: This paper achieves zero-shot generalization based solely on renderings and 3DGS features, unlike prior work 3DGS-Reloc[1], which relied on scene-specific LiDAR scans.
2. Feature fusion architecture is well-motivated. Outperforms single-path approaches in GS-CPR [2].

Weaknesses:
1. In this paper, there are 4096 selected 3DGS for training and inference. For large-scale scenes, this resolution may reduce relocation accuracy.
2. The method proposed in this paper may require higher gpu memory usage compared to other models.

[1] Jiang P, Pandey G, Saripalli S. 3dgs-reloc: 3d gaussian splatting for map representation and visual relocalization[J]. arXiv preprint arXiv:2403.11367, 2024.

[2] Liu C, Chen S, Bhalgat Y S, et al. GS-CPR: Efficient camera pose refinement via 3d gaussian splatting[C]//The Thirteenth International Conference on Learning Representations. 2025.

---

> ### Author Rebuttal · Authors · 2025-07-30
>
> **Responses to Reviewer 54m2**
>
> We express our sincere gratitude for your insightful suggestions and valuable feedback, which have significantly enhanced the quality of our manuscript.
>
> **Weaness 1**. Large-scale scenes with 4096 Gaussians.
>
> **Responses**. We sincerely appreciate the reviewer’s insightful comment. To address large-scale scenes, our framework may adopt a coarse-to-fine strategy. In the coarse stage, we uniformly sample 3D Gaussians across the entire scene and select those with high confidence scores, iterating this process as needed. In the refinement stage, we focus on Gaussians in proximity to the high-confidence selections, establishing 2D-3D correspondences between image pixels and these 3D Gaussians, followed by PnP to estimate the camera pose.
>
> **Weaness 2**. GPU memory usage.
>
> **Responses**. We appreciate the reviewer’s insightful query regarding GPU memory usage. As the two referenced methods do not report GPU memory details, a direct comparison is not feasible. In our framework, model features for the 3DGS branch are pre-computed prior to relocalization, allowing us to execute only the RGB branch and the feature fusion module during inference. This approach significantly reduces GPU memory consumption. Additionally, we report the model parameters: GS-RelocNet comprises approximately 81M parameters, while the pose refinement network contains 12M parameters.
>
> **Question 1**. Different scales and sizes of 3D Gaussians..
>
> **Responses**. We sincerely appreciate the reviewer’s insightful comment. Our relocalization framework establishes 2D-3D correspondences between image pixels and 3D Gaussians, followed by PnP to estimate the camera pose. For each 3D Gaussian, we employ confidence regression to identify its most closely matched pixel. The scale and size of 3D Gaussians are represented by their covariance matrix, which is incorporated into the loss function (Equation 1). This scale and size information enhances the accuracy of confidence regression.
>
> For the sampling strategy, we uniformly sample 3D Gaussians within grid cells to ensure a sufficient number of correspondences for robust pose estimation. Within each grid cell, Gaussians are randomly selected for training. Although we use 4096 Gaussians per training iteration, multiple iterations allow most Gaussians to be utilized. To evaluate the robustness of this strategy, we conducted experiments using random sampling instead of uniform grid-based sampling. The results show mean pose errors of 0.76cm/ 0.26° in the scene-dependent setting on the 7-Scenes dataset and 7cm/ 0.18° on the Cambridge Landmarks dataset. These results indicate minimal performance degradation, demonstrating the robustness of our sampling strategy.
>
> **Question 2**. Environment conditions.
>
> **Responses**. We sincerely appreciate the reviewer’s insightful query regarding environmental conditions. Our current implementation does not include specific adaptations for environmental changes, such as variations in lighting or weather. Upon reviewing other 3DGS based methods, including GS-CPR and STDLoc, we found no explicit mention of specialized designs addressing such conditions. To mitigate the impact of environmental variations, we propose exploring 3DGS variants designed for enhanced robustness to environmental changes, as demonstrated in recent works [1, 2]. These approaches could be integrated into our framework to improve performance under diverse conditions.
>
> [1] Zhang D, Wang C, Wang W, et al. Gaussian in the wild: 3d gaussian splatting for unconstrained image collections. ECCV 2024.
>
> [2] Kulhanek J, Peng S, Kukelova Z, et al. Wildgaussians: 3d gaussian splatting in the wild. arXiv preprint arXiv:2407.08447, 2024.
>
> **Question 3**. Limited datasets.
>
> **Responses**. Thanks for the helpful comment.
>  To further show the generalization capabilities of GS-RelocNet, we make more experiments as suggested. Specifically, we conducted additional experiments as recommended, evaluating GS-RelocNet on two challenging indoor dynamic datasets: TUM RGB-D and Bonn. Notably, our training data from ScanNet consist exclusively of static scenes, while the test sequences include dynamic elements, posing a significant challenge for generalization.
>
> In the TUM RGB-D test sequences, dynamic scenarios involving two individuals walking around a table increase the complexity of localization. Similarly, the Bonn dataset features highly dynamic sequences, such as individuals manipulating boxes or interacting with balloons. These experiments demonstrate the robustness of GS-RelocNet, which achieves substantial improvements in localization accuracy compared to existing SLAM systems.
>
> Table 1. RMSE of ATE [cm] results in four dynamic scenes of TUM RGB-D.
>
> | | fr3_walking_xyz | fr3_walking_static | fr3_walking_rpy | fr3_walking_half |  Mean |
> | :------: | :------: |:------: |:------: |:------: |:------: |
> | ORB-SLAM2 [1] | 45.9 | 9.3 | 65.8 | 32.8 | 38.5 |
> | DynaSLAM [2]  | 1.5 | 0.6 | 3.5 | 2.5 | 2.0 |
> | DS-SLAM [3]   | 2.5 | 0.8  | 44.4 | 3.0  | 12.7 |
> | LC-CRF SLAM [4] | 1.6 | 1.1 | 4.6 | 2.8 | 2.5 |
> | DGM-VINS [5] | 3.6 | 1.3 |7.1 | 3.3 | 3.8 |
> | Ours | **1.1** | **0.4** | **2.2** | **2.0**| **1.4**|
>
> **RMSE of ATE on dynamic TUM RGB-D**.
> Table 1 presents the Root Mean Square Error (RMSE) of Absolute Trajectory Error (ATE) for four dynamic scenes, compared against ORB-SLAM2, DynaSLAM, DS-SLAM, LC-CRF SLAM, and DGM-VINS. Our approach consistently outperforms these state-of-the-art SLAM systems in the RMSE of ATE metric, demonstrating superior localization accuracy in dynamic environments.
>
> **RMSE of ATE on dynamic Bonn**.
> We evaluated our framework on the Bonn dataset across 20 test scenes, consistent with LC-CRF SLAM, and compared it with ReFusion [6], MaskFusion [7], and LC-CRF SLAM. The mean RMSE of ATE results are 23.8cm (ReFusion), 25.1cm (MaskFusion), 6.8cm (LC-CRF SLAM), and 4.3cm (Ours). Our framework achieves the lowest mean RMSE of ATE, highlighting its exceptional localization accuracy in highly dynamic scenes involving objects such as manipulated boxes or balloons.
>
> **Generalization on Cambridge Landmarks**.
> To assess generalization, we trained GS-RelocNet on the MegaDepth dataset [8]  and tested it on the Cambridge Landmarks dataset in a scene-independent setting. The mean pose error across four scenes is 9cm/0.21°, surpassing the performance of DUSt3R (10cm/0.26°) and Reloc3R (38cm/0.49°). These results underscore the robustness and generalization capability of GS-RelocNet in diverse, unseen environments.
>
> **Failure case discussion**. In the Stairs scene of the 7-Scenes dataset, our framework exhibits failures in regions with repetitive or low-texture patterns. These failures likely stem from the visual ambiguity caused by similar views across different positions, which confounds accurate camera pose estimation using solely RGB inputs. To address this limitation, incorporating temporal sequence information could enhance robustness by providing additional context for disambiguating such challenging regions.
>
> **Reference**
>
> [1] R. Mur-Artal and J. D. Tardos, “Orb-slam2: An open-source slam ´
> system for monocular, stereo, and rgb-d cameras,” IEEE Transactions
> on Robotics, vol. 33, no. 5, pp. 1255–1262, 2017.
>
> [2] B. Bescos, J. M. Facil, J. Civera, and J. Neira, “Dynaslam: Tracking, ´
> mapping, and inpainting in dynamic scenes,” IEEE Robotics and Automation Letters, vol. 3, no. 4, pp. 4076–4083, 2018
>
> [3] C. Yu, Z. Liu, X.-J. Liu, F. Xie, Y. Yang, Q. Wei, and Q. Fei, “Ds-slam:
> A semantic visual slam towards dynamic environments,” in IEEE/RSJ
> International Conference on Intelligent Robots and Systems, 2018, pp.
> 1168–1174.
>
> [4] Z.-J. Du, S.-S. Huang, T.-J. Mu, Q. Zhao, R. R. Martin, and K. Xu,
> “Accurate dynamic slam using crf-based long-term consistency,” IEEE
> Transactions on Visualization and Computer Graphics, vol. 28, no. 4,
> pp. 1745–1757, 2020
>
> [5] B. Song, X. Yuan, Z. Ying, B. Yang, Y. Song, and F. Zhou, “Dgmvins: Visual-inertial slam for complex dynamic environments with
> joint geometry feature extraction and multiple object tracking,” IEEE
> Transactions on Instrumentation and Measurement, vol. 72, pp. 1–11,
> 2023.
>
> [6] E. Palazzolo, J. Behley, P. Lottes, P. Giguere, and C. Stachniss, “Refusion: 3d reconstruction in dynamic environments for rgb-d cameras
> exploiting residuals,” in IEEE/RSJ International Conference on Intelligent Robots and Systems, 2019, pp. 7855–7862.
>
> [7] M. Runz, M. Buffier, and L. Agapito, “Maskfusion: Real-time recognition, tracking and reconstruction of multiple moving objects,” in IEEE
> International Symposium on Mixed and Augmented Reality, 2018, pp.
> 10–20.
>
> [8] Zhengqi Li and Noah Snavely. Megadepth: Learning single-view depth prediction from internet photos.  CVPR 2018.

---

> > ### Comment · Reviewer_54m2 · 2025-08-05
> >
> > Thanks the authors for the detailed response. I will keep my rating positive given most of my concerns are well addressed.

---

### Official Review · Reviewer_EzZn · 2025-06-20

**Clarity:** 2
**Significance:** 3
**Originality:** 3
**Rating:** 4
**Confidence:** 4

**Summary:**

This paper proposes a novel scene-independent camera relocalization framework based on 3D GS, comprising two stages: initial pose estimation via 2D-3D correspondences between image pixels and 3D Gaussians, and pose refinement using rendered images from 3D GS. The framework introduces GS-RelocNet to establish correspondences and integrates a unidirectional 2D-3D feature fusion module and a bidirectional image feature fusion module. Experiments on 7 Scenes and Cambridge Landmarks demonstrate state-of-the-art performance, particularly in scene-independent relocalization, with significant improvements over existing methods.

**Questions:**

1. It is suggested to evaluate the generalization performance of the proposed method across multiple test datasets used for scene-independent methods, similar to how methods like DUSt3R and VGGT are evaluated.

2. It is suggested to briefly explain the connections of each method module in the caption of Figure 1 to enhance readability.

3. There is a contradiction between "Scene-independent" and "Relocalization" in the title, as Relocalization inherently implies scene-dependent settings. It is suggested to replace "Relocalization" with "Localization".

**Ethical Concerns:**

["NO or VERY MINOR ethics concerns only"]

**Final Justification:**

The authors have addressed my concerns. It is impressive that the proposed method achieves better results than DUSt3R despite using significantly less training data. Therefore, I am increasing my score to 4.

**Limitations:**

The experimental scenes in the paper are relatively simple and structured. It is suggested to enhance the method's performance in large-scale scenes with extreme appearance variations, which are more significant for real-world applications.

**Paper Formatting Concerns:**

The manuscript follows all formatting guidelines.

**Quality:**

3

**Strengths And Weaknesses:**

Strengths
1. The first scene-independent relocalization framework using 3D GS.

2. The unidirectional and bidirectional fusion modules enhance feature sharing, validated by ablation studies showing substantial performance gains.

3. Runs at 15.4 FPS, outperforming other 3D GS-based methods, making it suitable for real-world applications.

Weaknesses
1. The paper emphasizes scene-independent methods. However, the paper only includes one benchmark on 7Scenes under the scene-independent setting. Therefore, more experiments (such as Cambridge Landmarks) under scene-independent conditions should be included.

2. The evaluated datasets are relatively simple and structured, where the current method already demonstrates excellent performance. It is suggested to conduct evaluations on more challenging datasets.

2. The Introduction is somewhat confusing. It is suggested to refine it to clearly express the logical links among the challenges, motivation, and methodology.

---

> ### Author Rebuttal · Authors · 2025-07-30
>
> **Responses to Reviewer EzZn**
>
> We express our sincere gratitude for your insightful suggestions and valuable feedback, which have significantly enhanced the quality of our manuscript.
>
> **Weanesses 1,2 & Question 1**. Generalization capabilities.
>
> **Responses**. Thanks for the helpful comment.
>  To show the generalization capabilities of GS-RelocNet, we make more experiments as suggested. Specifically, we conducted additional experiments as recommended, evaluating GS-RelocNet on two challenging indoor dynamic datasets: TUM RGB-D and Bonn. Notably, our training data from ScanNet consist exclusively of static scenes, while the test sequences include dynamic elements, posing a significant challenge for generalization.
>
> In the TUM RGB-D test sequences, dynamic scenarios involving two individuals walking around a table increase the complexity of localization. Similarly, the Bonn dataset features highly dynamic sequences, such as individuals manipulating boxes or interacting with balloons. These experiments demonstrate the robustness of GS-RelocNet, which achieves substantial improvements in localization accuracy compared to existing SLAM systems.
>
> Table 1. RMSE of ATE [cm] results in four dynamic scenes of TUM RGB-D.
>
> | | fr3_walking_xyz | fr3_walking_static | fr3_walking_rpy | fr3_walking_half |  Mean |
> | :------: | :------: |:------: |:------: |:------: |:------: |
> | ORB-SLAM2 [1] | 45.9 | 9.3 | 65.8 | 32.8 | 38.5 |
> | DynaSLAM [2]  | 1.5 | 0.6 | 3.5 | 2.5 | 2.0 |
> | DS-SLAM [3]   | 2.5 | 0.8  | 44.4 | 3.0  | 12.7 |
> | LC-CRF SLAM [4] | 1.6 | 1.1 | 4.6 | 2.8 | 2.5 |
> | DGM-VINS [5] | 3.6 | 1.3 |7.1 | 3.3 | 3.8 |
> | Ours | **1.1** | **0.4** | **2.2** | **2.0**| **1.4**|
>
> **RMSE of ATE on dynamic TUM RGB-D**.
> Table 1 presents the Root Mean Square Error (RMSE) of Absolute Trajectory Error (ATE) for four dynamic scenes, compared against ORB-SLAM2, DynaSLAM, DS-SLAM, LC-CRF SLAM, and DGM-VINS. Our approach consistently outperforms these state-of-the-art SLAM systems in the RMSE of ATE metric, demonstrating superior localization accuracy in dynamic environments.
>
> **RMSE of ATE on dynamic Bonn**.
> We evaluated our framework on the Bonn dataset across 20 test scenes, consistent with LC-CRF SLAM, and compared it with ReFusion [6], MaskFusion [7], and LC-CRF SLAM. The mean RMSE of ATE results are 23.8cm (ReFusion), 25.1cm (MaskFusion), 6.8cm (LC-CRF SLAM), and 4.3cm (Ours). Our framework achieves the lowest mean RMSE of ATE, highlighting its exceptional localization accuracy in highly dynamic scenes involving objects such as manipulated boxes or balloons.
>
> **Generalization on Cambridge Landmarks**.
> To assess generalization, we trained GS-RelocNet on the MegaDepth dataset [8]  and tested it on the Cambridge Landmarks dataset in a scene-independent setting. The mean pose error across four scenes is 9cm/0.21°, surpassing the performance of DUSt3R (10cm/0.26°) and Reloc3R (38cm/0.49°). These results underscore the robustness and generalization capability of GS-RelocNet in diverse, unseen environments.
>
> **Reference**
>
> [1] R. Mur-Artal and J. D. Tardos, “Orb-slam2: An open-source slam ´
> system for monocular, stereo, and rgb-d cameras,” IEEE Transactions
> on Robotics, vol. 33, no. 5, pp. 1255–1262, 2017.
>
> [2] B. Bescos, J. M. Facil, J. Civera, and J. Neira, “Dynaslam: Tracking, ´
> mapping, and inpainting in dynamic scenes,” IEEE Robotics and Automation Letters, vol. 3, no. 4, pp. 4076–4083, 2018
>
> [3] C. Yu, Z. Liu, X.-J. Liu, F. Xie, Y. Yang, Q. Wei, and Q. Fei, “Ds-slam:
> A semantic visual slam towards dynamic environments,” in IEEE/RSJ
> International Conference on Intelligent Robots and Systems, 2018, pp.
> 1168–1174.
>
> [4] Z.-J. Du, S.-S. Huang, T.-J. Mu, Q. Zhao, R. R. Martin, and K. Xu,
> “Accurate dynamic slam using crf-based long-term consistency,” IEEE
> Transactions on Visualization and Computer Graphics, vol. 28, no. 4,
> pp. 1745–1757, 2020
>
> [5] B. Song, X. Yuan, Z. Ying, B. Yang, Y. Song, and F. Zhou, “Dgmvins: Visual-inertial slam for complex dynamic environments with
> joint geometry feature extraction and multiple object tracking,” IEEE
> Transactions on Instrumentation and Measurement, vol. 72, pp. 1–11,
> 2023.
>
> [6] E. Palazzolo, J. Behley, P. Lottes, P. Giguere, and C. Stachniss, “Refusion: 3d reconstruction in dynamic environments for rgb-d cameras
> exploiting residuals,” in IEEE/RSJ International Conference on Intelligent Robots and Systems, 2019, pp. 7855–7862.
>
> [7] M. Runz, M. Buffier, and L. Agapito, “Maskfusion: Real-time recognition, tracking and reconstruction of multiple moving objects,” in IEEE
> International Symposium on Mixed and Augmented Reality, 2018, pp.
> 10–20.
>
> [8] Zhengqi Li and Noah Snavely. Megadepth: Learning single-view depth prediction from internet photos.  CVPR 2018.
>
> **Weaness 3**. Improvements to the introduction.
>
> **Responses**. We sincerely thank the reviewer for their valuable feedback. Below, we outline the proposed enhancements to the introduction, addressing challenges, motivation, and methodology.
>
> **Challenges**. The 3DGS model excels in novel view synthesis, offering new opportunities for visual localization pipelines. However, leveraging 3DGS for robust and accurate localization remains a significant challenge.
>
> **Motivation**. Current methods predominantly utilize 3D Gaussians for pose refinement, which heavily relies on the accuracy of initial camera pose estimation. When the initial pose estimation is inaccurate or fails, the refinement process becomes ineffective. To address this, our work proposes a novel framework that integrates 3DGS for both pose initialization and refinement, while achieving scene-independent relocalization to enhance robustness.
>
> **Methodology**. We introduce a novel relocalization framework that first establishes 2D-3D correspondences between image pixels and 3D Gaussians, followed by a refinement stage that predicts the relative pose between real and rendered views using 3DGS. A feature fusion module is incorporated in both stages to enhance correspondence regression. To the best of our knowledge, this is the first 3DGS-based, scene-independent relocalization framework, offering a robust solution for challenging localization tasks.
>
> **Question 2**. Connections between method modules in the caption of Figure 1.
>
> **Responses**. We sincerely appreciate the reviewer’s insightful comment. The revised caption for Figure 1 is presented below.
>
> Architecture of GS-RelocNet. The GS-RelocNet framework integrates an RGB feature encoder, a 3D Gaussian feature encoder, an RGB descriptor decoder, a point descriptor decoder, and a confidence metric regression module. In the figure, arrows indicate the process flow, and numbers adjacent to each block denote the corresponding filter size.
>
> **Question 3**. "Scene-independent" and "Relocalization".
>
> **Responses**. We sincerely appreciate the reviewer’s insightful comment. Our method performs relocalization by regressing the camera pose within the coordinate space of a known 3DGS model, aligning with the standard definition of relocalization. The term "scene-independent" indicates that our framework can achieve robust relocalization in a target scene without requiring scene-specific pre-training, in contrast to most "scene-dependent" methods that necessitate prior training on the target scene.
>
> In the revised version, we add these to express the terms more clearly.

---

> ### Comment · Reviewer_EzZn · 2025-08-04
>
> Thank you to the authors for the detailed rebuttal. You have addressed my concerns. It is impressive that your method achieves better results than DUSt3R despite using significantly less training data. Therefore, I am increasing my score to 4. Moreover, I encourage the authors to improve the introduction to enhance readability and better motivate the work.

---

> > ### Author Response · Authors · 2025-08-05
> > **Official Comment by Authors**
> >
> > Thanks for your thoughtful feedback and score rasing. Your suggestions are invaluable in helping us improve the paper. We sincerely appreciate your recognition of our efforts to address the major concerns. We will thoroughly revise the paper to incorporate all of your feedback.

---

### Official Review · Reviewer_TpKv · 2025-06-30

**Clarity:** 2
**Significance:** 3
**Originality:** 2
**Rating:** 4
**Confidence:** 4

**Summary:**

The paper proposes a camera localization method based on 3D Gaussian splatting.

In the first stage, the method generates 2D–3D correspondences. A neural network, GS-RelocNet, takes as input the RGB image and the 3D Gaussians, and outputs a confidence matrix. 2D–3D correspondences are derived from this matrix, and the pose is then estimated using PnP+RANSAC. Within GS-RelocNet, a unidirectional feature fusion module is proposed. GS-RelocNet is trained to regress a ground-truth confidence matrix, where each entry encodes the Mahalanobis distance between pixel coordinates and the reprojected 3D points.


The second stage is a refinement step. A refinement network takes the real RGB image and a rendered RGB image and outputs a residual coordinate map, following Wang & Qi. Here, a bidirectional feature fusion module is proposed. The network is trained using a coordinate loss map and a pose loss.

The approach is evaluated on 7-Scenes, Cambridge Landmarks, and an additional dataset (12-Scenes, in the supplementary material).
In the scene-dependent setting, the results are comparable to the state-of-the-art STDLoc (CVPR 2025).
In the scene-independent setting, the proposed method is the first to use Gaussian splatting and significantly outperforms existing methods on 7-Scenes.
Moreover, the method is considerably faster than its competitors: 65 ms versus 143 ms for STDLoc.

**Questions:**

1. **Ablation study** -  Could you include an architecture using cross-attention layers in the ablation study, or explain why this was not considered appropriate?
2. **Technical details** - Since the code will not be released, could you provide the missing details (e.g., tensor dimensions) to make re-implementation easier?
3. **Auxiliary loss** - For the auxiliary loss, how did you backpropagate through the predicted camera pose produced by the RANSAC module?
4. **Computation time** - Could you provide a breakdown of the computation time for each module (GS-RelocNet, PnP+RANSAC, and refinement iterations)?

**Ethical Concerns:**

["NO or VERY MINOR ethics concerns only"]

**Final Justification:**

The authors have addressed most of my initial concerns (Ablation study, Technical details, Auxiliary loss and Computation time), and I have accordingly increased my rating.

**Limitations:**

Although the authors indicate in the checklist that limitations are discussed, I was unable to find any such discussion in the main paper or the supplementary material.

**Paper Formatting Concerns:**

/

**Quality:**

3

**Strengths And Weaknesses:**

## Strengths
1. The paper proposes the first scene-independent camera localization method based on 3D Gaussian splatting.
2. The method is computationally efficient.
3. The outperforms state-of-the-art methods on 7-Scenes in the scene-independent setting.
4. An extensive ablation study supports the architectural and design choices.

## Weaknesses
1. Only a single NeurIPS paper (Zhu, 2024) is cited, which raises concerns about whether the paper is fully within the scope of NeurIPS.

2. The authors do not plan to release their code and provide no justification. Reproducing the results may be challenging, as some implementation details are missing. For example, in the description of the Unidirectional Feature Fusion module, it appears that RGB features are reshaped to apply the residual connection, but the input and output dimensions are not specified.

3. Two main contributions are the unidirectional and bidirectional feature fusion modules, but it is unclear why classical cross-attention layers were not considered instead. I would have expected a comparison with more standard cross-attention architectures (e.g., Dust3R’s ViT variant) in the ablation study.

---

> ### Author Rebuttal · Authors · 2025-07-30
>
> **Responses to Reviewer TpKv**
>
> We express our sincere gratitude for your insightful suggestions and valuable feedback, which have significantly enhanced the quality of our manuscript.
>
> **Weaness 1**. NeurIPS paper citation.
>
> **Responses**. Thanks for the helpful comment.
> We add the following papers in our paper.
>
> [1] Zhu J, Yan S, Wang L, et al. LoD-Loc: Aerial Visual Localization using LoD 3D Map with Neural Wireframe Alignment[J]. NIPS 2024.
>
> [2] Xu Y, Jiang H, Xiao Z, et al. Dg-slam: Robust dynamic gaussian splatting slam with hybrid pose optimization. NIPS 2024.
>
> [3] Hu J, Mao M, Bao H, et al. CP-SLAM: Collaborative neural point-based SLAM system. NIPS 2023.
>
> [4] Wang Y, Yan Y, Shi D, et al. NeRF-IBVS: visual servo based on nerf for visual localization and navigation. NIPS 2023.
>
> **Weaness 2 & Question 2**. Technical details.
>
> **Responses**. We appreciate the reviewer’s insightful comment and apologize for the lack of clarity in describing the unidirectional feature fusion module. The module is designed to maintain the same output size as the input RGB feature. Let the input RGB feature have dimensions $H_u \times W_u \times D_u$, and the model feature have dimensions $N_m \times D_m$. The fusion process proceeds as follows.
>
> 1.The RGB feature is reshaped to $N_m \times (H_u \times W_u \times D_u / N_m)$, followed by a 1D convolution to transform it to $N_m \times D_m$, aligning with the model feature’s dimensions.
>
> 2.Both features undergo self-attention operations and are subsequently combined through element-wise addition.
>
> 3.The combined feature is reshaped to $H_u \times W_u \times (N_m \times D_m / H_u / W_u)$. A Swin Transformer block is then applied to restore the feature dimensions to $H_u \times W_u \times D_u$, matching the input RGB feature. The input RGB feature is added to this output to produce the final merged feature.
>
> We acknowledge an error in the bottom left part of Figure 1, where a redundant line depicting the addition of the input RGB feature was included; this will be corrected in the revised manuscript. We apologize for this oversight.
>
> For details on the 3DGS and RANSAC implementations, please refer to **Weaness 4 & Question 2** of **Reviewer sxyi**.
>
> As stated in the submission checklist, the code is not publicly available during the review stage. **Upon publication, the core code will be released to ensure reproducibility.**
>
> **Weaness 3 & Question 1**. Ablation study of cross-attention layers.
>
> **Responses**. We appreciate the reviewer’s insightful comment regarding the efficacy of cross-attention layers, which have demonstrated strong performance in prior work. Our decision to exclude cross-attention layers in the unidirectional feature fusion module stems from the need to preserve the independence of the 3DGS branch. This design ensures that the point cloud features remain unaffected by RGB features during inference, thereby maintaining computational efficiency. Incorporating cross-attention layers would allow 3DGS features to influence RGB features, significantly increasing the framework’s runtime.
>
> To explore the potential of cross-attention layers, we conducted two ablation studies in the pose refinement stage. First, we integrated a cross-attention layer into the bidirectional feature fusion module of the refinement network. This yielded mean pose errors of 0.73 cm / 0.24° in the scene-dependent setting on the 7-Scenes dataset and 7cm/ 0.17 on the Cambridge Landmarks dataset, comparable to our original results. Second, we replaced the refinement network with Dust3R’s ViT variant, incorporating cross-attention. This resulted in mean pose errors of 0.76 cm / 0.27° on 7-Scenes and 9cm/ 0.21 on Cambridge Landmarks, indicating slightly reduced accuracy. We attribute the limited impact of cross-attention layers to the small input resolution of the refinement network ($128 \times 128$), which constrains their effectiveness.
>
> **Question 3**. Auxiliary loss.
>
> **Responses**.  We appreciate the reviewer’s query regarding the auxiliary loss. In our approach, the auxiliary loss directly predicts the 6-DoF pose (3-DoF translation and 3-DoF rotation) rather than estimating the pose via PnP from coordinates, similar to the methodology in PoseNet [1]. This loss is designed to enhance the geometric awareness of the pose refinement network, improving its ability to capture spatial relationships effectively.
>
> [1] Kendall, Alex and Cipolla, Roberto. Geometric loss functions for camera pose regression with deep learning. CVPR 2017
>
> **Question 4**. Computation time.
>
> **Responses**. Our method comprises five key steps, including confidence regression, initial pose estimation using PnP with RANSAC, view rendering, residual coordinate regression, and pose refinement using PnP with RANSAC. We evaluated the computational efficiency of these steps across all test frames in the 7-Scenes and Cambridge Landmarks datasets. The average running times per frame are as follows: 39 ms for confidence regression, 4 ms for initial PnP with RANSAC, 9 ms for view rendering, 8 ms for residual coordinate regression, and 5 ms for PnP in pose refinement.
>
> **Limitation discussion**. A primary limitation of our framework is its reliance on a high-quality 3D GS model of the target scene. When the 3D GS model is of suboptimal quality, localization performance may degrade, leading to failures or significant errors.

---

> > ### Comment · Reviewer_TpKv · 2025-08-02
> >
> > Thank you to the authors for the detailed rebuttal. You have addressed my questions clearly, and I have no further questions or points to discuss.

---

> > > ### Author Response · Authors · 2025-08-05
> > > **Official Comment by Authors**
> > >
> > > We're glad our rebuttal has addressed your questions. Thank you again for reviewing our work and providing invaluable suggestions!

---

### Official Review · Reviewer_sxyi · 2025-07-02

**Clarity:** 3
**Significance:** 3
**Originality:** 3
**Rating:** 5
**Confidence:** 5

**Summary:**

The paper tackles the task visual localization which consists in estimating the 6DoF pose of a query image within a known environment. It is a crucial task for AR/MR and autonomous-driving systems. One popular way of solving pose estimation is regressing the 3D position (in world coordinates) of each pixel within an image and estimating the pose through RANSAC+PnP. However this requires fitting the regression network on each scene independently and the underlying 3D point cloud representation lacks dense texture information. The authors hence leverage 3DGS representations which jointly encodes scene geometry and texture by introducing a scene independent visual localization pipeline. It consists of GS-RelocNet which establishes pixel to 3D Gaussians correspondences (used with a PnP+RANSAC loop to estimate an initial pose) and a refinement network that regresses residual coordinate maps between the query image and the image rendered from the initial pose (used to update the initial pose by estimating the relative pose with PnP+RANSAC). Both modules use fusion mechanisms to share information between different modalities in their two branches. The resulting pipeline is evaluated on two real world standard relocalization datasets.

**Questions:**

1) Do you have additional results to evaluate the generalization capabilities of GS-RelocNet?

2) Which underlying 3DGS model is used? Which hyperparameters in the 3DGS model? How are the point clouds used to initialize the 3DGS models obtained?
There are some illumination variations in Cambridge Landmark, is this explicitly handled somewhere in the pipeline?

3) For a query image, could you break down the runtime between the different steps of your pipeline?

4) How many descriptors Ni per image or what is the patch embedding size?

Additional references for 3DGS visual localization:
Gennady Sidorov, Malik Mohrat, Ksenia Lebedeva, Ruslan Rakhimov, and Sergey Kolyubin. “GSplatLoc: Grounding Keypoint Descriptors into 3D Gaussian Splatting for Improved Visual Localization.”
Pietrantoni, Maxime, Gabriela Csurka, and Torsten Sattler. "Gaussian Splatting Feature Fields for (Privacy-Preserving) Visual Localization." In CVPR 2025
Hongjia Zhai, Xiyu Zhang, Boming Zhao, Hai Li, Yijia He, Zhaopeng Cui, Hujun Bao, and Guofeng Zhang. “SplatLoc: 3D Gaussian Splatting-based Visual Localization for Augmented Reality.” In TVCG 2025

**Ethical Concerns:**

["NO or VERY MINOR ethics concerns only"]

**Final Justification:**

My concerns have been addressed through the rebuttal. Given the novelty of the approach and the state of the art localization accuracy I see no reason to reject the paper, I have upgraded my score accordingly.

**Limitations:**

Limitations not addressed

**Quality:**

3

**Strengths And Weaknesses:**

### Strengths
1) Conceptually, learning to regress point to 3D Gaussian correspondences is a good intermediate solution between feature matching and scene coordinate regression. GS-RelocNet combines two powerful backbones for image and point clouds through a feature fusion module which allows it to predict such correspondences with high accuracy. It is further supported by an ablation study.
2) 3DGS scene independent visual localization is a very relevant research direction and GS-RelocNet generalizes to unseen indoor scenes during training yielding very accurate pose estimation.
3) The relocalization pipeline achieves state of the art results against rendering-based baselines and regression baselines on two standard relocalization datasets.
4) The feedforward regression nature of the method makes the inference very efficient and allows the relocalization localization pipeline to almost run at real time.

### Weaknesses
1) Given that the scene-independent nature of the pipeline is advertised as the main contribution of the paper I find experiments around generalization to be too limited. Notably, the scene independent version is evaluated only in indoor environments which inherently facilitates generalization as room layouts and the semantic content of rooms are relatively similar between scenes. Furthermore, 7-Scenes is not a challenging dataset for visual localization given the small scale of the scenes and the dense distribution of training cameras.

2) l.466 it is mentioned that for 7-Scenes, “ground truth poses were obtained using the KinectFusion system”. However, it seems like your pipeline (and some other methods in Tab. 1) is evaluated against the SfM pseudo ground truth poses from [1] while the rest of the methods in the table report results evaluated against the original Dslam pseudo ground truth poses. It is important that all methods within the table are evaluated against the same pseudo ground truth as results can significantly vary between both ground truths. Could you clarify which pseudo ground truth was used and if necessary evaluate your method against both ground truths.

[1] On the Limits of Pseudo Ground Truth in Visual Camera Re-Localisation. Eric Brachmann, Martin Humenberger, Carsten Rother, Torsten Sattler

3) Matching based structure based methods ought to at least be mentioned in the introduction as, overall, they remain the visual localization approach with the best robustness and accuracy on real world scenes.

[2] Paul-Edouard Sarlin, Cesar Cadena, Roland Siegwart, and Marcin Dymczyk. From Coarse to Fine: Robust Hierarchical Localization at Large Scale. In CVPR, 2019.

[3] Martin Humenberger, Yohann Cabon, Nicolas Guerin, Julien Morat, Jérôme Revaud, Philippe Rerole, Noé Pion, César Roberto de Souza, Vincent Leroy and Gabriela Csurka. Robust Image Retrieval-based Visual Localization using Kapture.

4) There is a total lack of information regarding the underlying 3DGS model.
Similarly there is no information regarding RANSAC hyperparameters (number of iterations, thresholds) which can heavily influence the runtime.
For reproducibility purposes, this information should be added in the supplementary material.

---

> ### Author Rebuttal · Authors · 2025-07-30
>
> **Responses to Reviewer sxyi**
>
> We express our sincere gratitude for your insightful suggestions and valuable feedback, which have significantly enhanced the quality of our manuscript.
>
> **Weaness 1 & Question 1**. Generalization capabilities of GS-RelocNet.
>
> **Responses**. The reviewer raises a critical concern regarding the generalization capabilities of GS-RelocNet, an essential aspect of relocalization tasks. We appreciate this insightful comment. To address this, we conducted additional experiments as recommended, evaluating GS-RelocNet on two challenging indoor dynamic datasets: TUM RGB-D and Bonn. Notably, our training data from ScanNet consist exclusively of static scenes, while the test sequences include dynamic elements, posing a significant challenge for generalization.
>
> In the TUM RGB-D test sequences, dynamic scenarios involving two individuals walking around a table increase the complexity of localization. Similarly, the Bonn dataset features highly dynamic sequences, such as individuals manipulating boxes or interacting with balloons. These experiments demonstrate the robustness of GS-RelocNet, which achieves substantial improvements in localization accuracy compared to existing SLAM systems.
>
> Table 1. RMSE of ATE [cm] results in four dynamic scenes of TUM RGB-D.
>
> | | fr3_walking_xyz | fr3_walking_static | fr3_walking_rpy | fr3_walking_half |  Mean |
> | :------: | :------: |:------: |:------: |:------: |:------: |
> | ORB-SLAM2 [1] | 45.9 | 9.3 | 65.8 | 32.8 | 38.5 |
> | DynaSLAM [2]  | 1.5 | 0.6 | 3.5 | 2.5 | 2.0 |
> | DS-SLAM [3]   | 2.5 | 0.8  | 44.4 | 3.0  | 12.7 |
> | LC-CRF SLAM [4] | 1.6 | 1.1 | 4.6 | 2.8 | 2.5 |
> | DGM-VINS [5] | 3.6 | 1.3 |7.1 | 3.3 | 3.8 |
> | Ours | **1.1** | **0.4** | **2.2** | **2.0**| **1.4**|
>
> **RMSE of ATE on dynamic TUM RGB-D**.
> Table 1 presents the Root Mean Square Error (RMSE) of Absolute Trajectory Error (ATE) for four dynamic scenes, compared against ORB-SLAM2, DynaSLAM, DS-SLAM, LC-CRF SLAM, and DGM-VINS. Our approach consistently outperforms these state-of-the-art SLAM systems in the RMSE of ATE metric, demonstrating superior localization accuracy in dynamic environments.
>
> **RMSE of ATE on dynamic Bonn**.
> We evaluated our framework on the Bonn dataset across 20 test scenes, consistent with LC-CRF SLAM, and compared it with ReFusion [6], MaskFusion [7], and LC-CRF SLAM. The mean RMSE of ATE results are 23.8cm (ReFusion), 25.1cm (MaskFusion), 6.8cm (LC-CRF SLAM), and 4.3cm (Ours). Our framework achieves the lowest mean RMSE of ATE, highlighting its exceptional localization accuracy in highly dynamic scenes involving objects such as manipulated boxes or balloons.
>
> **Generalization on Cambridge Landmarks**.
> To assess generalization, we trained GS-RelocNet on the MegaDepth dataset [8]  and tested it on the Cambridge Landmarks dataset in a scene-independent setting. The mean pose error across four scenes is 9cm/0.21°, surpassing the performance of DUSt3R (10cm/0.26°) and Reloc3R (38cm/0.49°). These results underscore the robustness and generalization capability of GS-RelocNet in diverse, unseen environments.
>
> **Reference**
>
> [1] R. Mur-Artal and J. D. Tardos, “Orb-slam2: An open-source slam ´
> system for monocular, stereo, and rgb-d cameras,” IEEE Transactions
> on Robotics, vol. 33, no. 5, pp. 1255–1262, 2017.
>
> [2] B. Bescos, J. M. Facil, J. Civera, and J. Neira, “Dynaslam: Tracking, ´
> mapping, and inpainting in dynamic scenes,” IEEE Robotics and Automation Letters, vol. 3, no. 4, pp. 4076–4083, 2018
>
> [3] C. Yu, Z. Liu, X.-J. Liu, F. Xie, Y. Yang, Q. Wei, and Q. Fei, “Ds-slam:
> A semantic visual slam towards dynamic environments,” in IEEE/RSJ
> International Conference on Intelligent Robots and Systems, 2018, pp.
> 1168–1174.
>
> [4] Z.-J. Du, S.-S. Huang, T.-J. Mu, Q. Zhao, R. R. Martin, and K. Xu,
> “Accurate dynamic slam using crf-based long-term consistency,” IEEE
> Transactions on Visualization and Computer Graphics, vol. 28, no. 4,
> pp. 1745–1757, 2020
>
> [5] B. Song, X. Yuan, Z. Ying, B. Yang, Y. Song, and F. Zhou, “Dgmvins: Visual-inertial slam for complex dynamic environments with
> joint geometry feature extraction and multiple object tracking,” IEEE
> Transactions on Instrumentation and Measurement, vol. 72, pp. 1–11,
> 2023.
>
> [6] E. Palazzolo, J. Behley, P. Lottes, P. Giguere, and C. Stachniss, “Refusion: 3d reconstruction in dynamic environments for rgb-d cameras
> exploiting residuals,” in IEEE/RSJ International Conference on Intelligent Robots and Systems, 2019, pp. 7855–7862.
>
> [7] M. Runz, M. Buffier, and L. Agapito, “Maskfusion: Real-time recognition, tracking and reconstruction of multiple moving objects,” in IEEE
> International Symposium on Mixed and Augmented Reality, 2018, pp.
> 10–20.
>
> [8] Zhengqi Li and Noah Snavely. Megadepth: Learning single-view depth prediction from internet photos.  CVPR 2018.
>
> **Weaness 2**. Ground truth of 7-Scenes dataset.
>
> **Responses**. We appreciate the reviewer’s insightful comment regarding the ground truth poses for the 7-Scenes dataset. As noted, the dataset provides two types of ground truth: the original poses from KinectFusion and the SfM pseudo ground truth. We apologize for the lack of clarity in our initial submission. In the revised manuscript, we explicitly specify the ground truth used for each method in Table 1 of the paper. Specifically, ACE, GLACE, GS-CPR, STDLoc and ours utilize the SfM pseudo ground truth, while other methods, leverage the original KinectFusion ground truth.
>
> To further address this concern, we conducted additional experiments using the original ground truth. Our method achieves mean pose errors of 1.21 cm / 0.61° in the scene-dependent setting and 1.44 cm / 0.82° in the scene-independent setting across the seven scenes. These results demonstrate superior accuracy compared to other methods evaluated with the original ground truth, highlighting the robustness of our approach.
>
> **Weaness 3**. Inclusion of match-based methods and references.
>
> **Responses**. We greatly appreciate the reviewer’s valuable suggestion. In response, we will incorporate a discussion of match-based methods into the introduction of the revised manuscript. Additionally, the referenced works have been included to provide a comprehensive background and context for our approach.
>
> **Weaness 4 & Question 2**. Implementation details of the 3DGS model and PnP with RANSAC.
>
> **Responses**. We sincerely appreciate the reviewer’s insightful comment. To construct the 3DGS model, we first utilize COLMAP to generate an initial point cloud using ground truth poses. Subsequently, we employ the original 3DGS model with its default configuration settings.
>
> --iterations: 30000. Total number of training iterations.
>
> --position\_lr\_init: 0.00016. The initial learning rate of the Gaussian position.
>
> --position\_lr\_final: 0.0000016. The final learning rate of the Gaussian position.
>
> --position\_lr\_delay\_mult: 0.01. The delay multiplier before the learning rate decay begins.
>
> --position\_lr\_max\_steps: 30000. The total number of steps for the learning rate decay.
>
> --feature\_lr: 0.0025. Learning rate of spherical harmonic function coefficients.
>
> --opacity\_lr: 0.05. Learning rate of opacity.
>
> --scaling\_lr: 0.005. Scaling learning rate.
>
> --rotation\_lr: 0.001. Learning rate of rotation.
>
> --densify\_from\_iter: 500. Densification begins from which iteration.
>
> --densify\_until\_iter: 15000. Densification ends at which iteration.
>
> --densification\_interval: 100. Perform densification and pruning checks every few iterations.
>
> --opacity\_prune\_threshold: 0.005. Opacity pruning threshold.
>
> --densify\_grad\_threshold: 0.0002. The gradient threshold for densifying the Gaussian sphere.
>
> The PnP with RANSAC uses OpenCV implementation with following parameters.
>
> --iterationsCount: 100. The number of RANSAC iterations.
>
> --reprojectionError: 8. Threshold for reprojection error.
>
> --confidence: 0.99. Degree of confidence.
>
> --flags:SOLVEPNP\_ITERATIVE. PnP solver algorithm.
>
> Regarding the code availability, we clarify in the submission checklist that the code is not publicly available during the review stage. Upon publication of the paper, the core code will be made publicly accessible to facilitate reproducibility.
>
> **Question 3**. Detailed running time of pipeline components.
>
> **Responses**. Our method comprises five key steps, including confidence regression, initial pose estimation using PnP with RANSAC, view rendering, residual coordinate regression, and pose refinement using PnP with RANSAC. We evaluated the computational efficiency of these steps across all test frames in the 7-Scenes and Cambridge Landmarks datasets. The average running times per frame are as follows: 39 ms for confidence regression, 4 ms for initial PnP with RANSAC, 9 ms for view rendering, 8 ms for residual coordinate regression, and 5 ms for PnP in pose refinement.
>
> **Question 4**. How many descriptors Ni per image or what is the patch embedding size?
>
> **Responses**. We thank the reviewer for this insightful question. The number of descriptors per image, $N_i$ per image is 4096 ($64 \times 64$).
> The patch embedding size of both GS-RelocNet and pose refinement network is $64 \times 64$.
>
> **Limitation discussion**. A primary limitation of our framework is its reliance on a high-quality 3D GS model of the target scene. When the 3D GS model is of suboptimal quality, localization performance may degrade, leading to failures or significant errors.

---

> > ### Comment · Reviewer_sxyi · 2025-08-04
> >
> > I thank the reviewers for their comprehensive rebutall. My points have been addressed and I have no further questions. Should the paper be accepted, I encourage the authors to include the additional generalization experiments as well as to ensure fair comparison on the 7-scenes dataset by separating both pseudo ground truths.

---

> > > ### Author Response · Authors · 2025-08-05
> > > **Official Comment by Authors**
> > >
> > > We're glad our rebuttal has addressed your points. The additional generalization experiments to ensure fair comparison on the 7-scenes dataset by separating both pseudo ground truths will added into the later version. Thank you again for reviewing our work and providing invaluable suggestions!

---

### Public Comment · ~David_Nordström1 · 2026-05-13
**Code not released**

Hi!

This paper has yet to release the code. However, it is explicitly said in the paper and the rebuttal that this would be done. The conference was appoximately 6 months ago.

Do you have a timeline for when the code will be released?

/ David

---

### Note · Authors · 2025-08-12

In the Final Remarks part, we would like to express our sincere gratitude for the insightful suggestions and constructive comments provided by all the reviewers. Their feedback has been instrumental in enhancing the quality of our manuscript.

In our rebuttal, we have addressed each concern and question raised by the reviewers in detail. Based on the subsequent official feedback from the reviewers, we are confident that the major concerns have been effectively resolved and approved. In the revised version of the manuscript, we have made comprehensive revisions, incorporating the valuable suggestions and comments to further improve the overall quality of the paper.

---

### Decision · Program_Chairs · 2025-09-17

**Decision:**

Accept (poster)

**Comment:**

This paper presents a novel two-stage, scene-independent visual localization framework using 3D Gaussian Splatting. It first establishes coarse 2D-3D correspondences for an initial pose estimate and then refines it using a rendered view, with both stages enhanced by specialized feature fusion modules. The proposed method demonstrate state-of-the-art accuracy on standard benchmarks. It is also highly efficient, operating at near real-time speeds.

After rebuttal, the final rates are 3 Borderline accepts and 1 Accept. The authors solved reviewers' concerns on generalization capabilities, computation overhead and ablation studies. The reviewers give positive comments and recognition. Given these positive reviews, the AC recommends acceptance and suggests to further improve the paper based on the reviewers' comments.